**Perspective**

# What defines a flood
**Mason Majszak** [1,2] ✉, **Abdallah Zaki** [3] & **Omar Wani** [1] ✉

While the term 'flood' may elicit clear images of rushing water or overflowing river banks, genuine concerns around formally demarcating this phenomenon remain. In this paper, we investigate the phenomenon's conceptual space, discussing its physical and social components. We highlight key features of the term, as it is commonly used, while illustrating existing conceptual challenges, including the lack of direct reference to the assumptions about how things ought to be, what the philosophical literature calls normativity, involved in setting relevance parameters and the rarity threshold. We propose defining the flood phenomenon by referencing both its physical and normative properties. Within this demarcation framework, physical floods constitute pools or fluxes of water beyond a normatively-specified rarity threshold. A second demarcation occurs when a notion of desirability is used to identify if the presence of water is bad or good. Anthropocentric floods are then physical floods deemed undesirable or desirable. Recognizing the uncertainty present in identifying and adopting notions of desirability can then help explain instances of lack of consensus in communities, what we call interventional ambiguity, on whether to seek engineered solutions to specific flood-related problems. Our demarcation framework can facilitate cross-disciplinary communication and motivate further refinement of flood-related concepts.

## Why the flood concept needs clearer demarcation criteria

Imagine you find yourself standing outside. You look down and see that you are in a knee-deep pool of water. As you continue to examine your surroundings, you see a number of plants, some cattails and willows. Our question to you—is this a flood? Some may say, "yes, my shoes are flooded", others may say, "no, this is a wetland, and this area normally has such a level of water". Take another example. While the region of a river that carries water year-round is categorized as part of the river channel, is its adjacent floodplain, which is inundated only periodically, a perennial part of the river system? It is often not considered that way and, therefore, may be seen as an attractive opportunity to develop expensive waterfront property. In order to meaningfully address these hypotheticals, we must ask a fundamental question—what is a flood?

While rarely defined explicitly within the hydroclimatology community, "flood" is generally used as a primitive notion on which other, second-order, definitions are built, such as "pluvial flood", "fluvial flood", "coastal flood", "urban flood", "flash flood", "debris flood", "nuisance flood", etc. Individuals often adopt their own working notion of what is or is not a flood, however, conceptually, there may be unarticulated disagreements. While, in general, it is not entirely necessary to provide precise definitions of primitive notions, conceptual agreement in the use of the term is helpful when constructing larger research frameworks. This need for conceptual clarity becomes a larger issue when discussing the broader use of the term and the impacts of the events in question.

In the face of climate change, compounded with growing populations, hydroclimatic hazards continue to be incredibly impactful for communities around the globe, with thousands of lives lost every year, millions displaced, and economic damages that reach into the hundreds of billions of dollars[1–3]. However, understanding the source of disagreements, where individuals may use the term "flood" to refer to different conceptualizations of the phenomenon, remains an outstanding challenge within the hydroclimatological, environmental, and disaster risk management communities.

In this paper, we argue that utilizing normative aspects, adopting a notion of the way things ought to or should be[4], is necessary for any conceptualization of "flood". These aspects are, generally, the evaluative judgments that underlie the decisions of what ought to be and necessarily involve utilizing values about desirable states of affairs[5]. Nevertheless, this reliance currently remains tacit; therefore, any comprehensive demarcation framework for the flood phenomenon, as we will argue, must explicitly recognize the role of this normativity.

## Current perspectives on floods

Much of the scientific discussion around floods focuses on the physical features. Take the Intergovernmental Panel on Climate Change (IPCC), they define "flood" as, "[t]he overflowing of the normal confines of a stream

[1]Tandon School of Engineering, New York University, Brooklyn, NY, USA. [2]Department of Philosophy, New York University, New York, NY, USA. [3]Department of Earth and Planetary Sciences, The University of Texas Jackson School of Geosciences, Austin, TX, USA. ✉e-mail: mason.majszak@nyu.edu; omarwani@nyu.edu

or other water body, or the accumulation of water over areas that are not normally submerged. Floods can be caused by unusually heavy rain, for example, during storms and cyclones. Floods include river (fluvial) floods, flash floods, urban floods, rain (pluvial) floods, sewer floods, coastal floods and glacial lake outburst floods"[6].

The IPCC's definition, and other similar definitions, does not elaborate on what constitutes "normal" and therefore treats floods as objective, referencing only specific physical attributes of the system when conceptualizing this phenomenon and its causes. These attributes include a description of the location of the event prior to the event taking place, such as the extent or level of water present at the location, and a description of potential causes of flooding, such as heavy rain. This physical account can be seen as the general working definition used by the hydroclimate community and can be examined from a number of relevant perspectives.

## How physical factors contribute to the conceptualization of floods

A physics-based perspective of floods can be made more precise and lends itself to a mathematical definition, where the set of floods $Q_r$ can be expressed as:

$$Q_r = \left\{ q \in \mathbb{R} \,|\, F(q) \geq 1 - 1/r \right\}$$

In this equation, $F(q)$ is the cumulative probability distribution of the magnitude of water-related pools or fluxes $q$, and $r$ is the expected return period, generally in years, associated with their exceedance. The return period refers to the average time it takes for a variable to exceed a threshold. $1/r$ is the exceedance probability. $Q_r$ then forms our notion of floods with respect to a certain distribution $F(q)$, determined by the geographic characteristics and hydroclimatic history of a location.

When looking at the hydroclimatology communities' working definition of "flood", this perspective provides insight on the concept of "normal" adopted in the definition and the appeal to the normal state of the system as one that is not submerged. While identifying what the normal conditions of the system are can be done statistically, there are methodological decisions that must be made, which impact what is considered normal. The conception of normal conditions is dependent on the identification of a relevant timescale and location of inquiry, as well as, crucially, a rarity threshold.

The rarity threshold, in this case the return period $r$, defines which events should be considered far enough on the tail of the distribution to no longer be considered normal. Notwithstanding, there is no single or well-defined threshold to set for the return period. In practice, return periods on the order of a few years to 100s of years are often utilized as they are the scale of concern for critical infrastructure and other features of concern relevant for communities[7]. The chosen return period is determined by the implicit or explicit level of risk tolerance of communities and individuals, and whether it leads to submergence of some specific locations. This demonstrates that the rarity threshold can be socially dictated or derived from a human notion of normativity, i.e., utilizing human-relevant scales or referencing things humans care about to define the way things ought to be.

The identification of normal conditions does not stop with defining the rarity threshold. Additionally, the identification of the relevant timescale and location of inquiry is also necessary. This identification is nontrivial—take, for example, geological timescales, within which the term "flood" encompasses a spectrum of events. This can include events from those that repeatedly cause river jumps and migration across floodplains on decadal, centennial, and millennial scales, to floods recorded in the sedimentary rock record spanning hundreds of thousands to millions of years of climatic change, and to catastrophic episodes triggered by threshold failures in storage-and-release systems rather than by fluctuations in a quasi-stationary hydroclimate. These end-member events differ from riverine floods in both mechanism and repeatability, and they are better treated as contingent "release" events than as realizations from a stationary time series. These events—especially the catastrophic ones—transport enormous volumes of

water and sediment, occur on timescales that far exceed human memory, and leave remarkable evidence on Earth's surface, differing substantially from modern-day notions of floods in both scale and frequency[8-10].

Floods also appear to have occurred on Mars and Titan in the past[9,11,12]. On those worlds, high-energetic flooding likely reshaped regional topography and reset boundary conditions for later surface evolution. While it can be argued that even extraterrestrial floods are classified as such because of their rarity, we focus on Earth, where floods operate within coupled feedbacks among water, rock, biota, and humans[13-15]. The imprint of such ancient floods is encoded in sedimentary and geomorphic archives, which record the interactions between dynamic surface processes and the extremes of Earth's environment. Planetwide reconstructions of Earth's past provide direct evidence that geological floods have profoundly shaped our planet's landscapes: from the erosion and deposition of large sedimentary bodies in central Spain during the paleocene-eocene thermal maximum[15], to the sculpting of the channeled scablands in North America[8], the separation of Britain from France via the English Channel[14], and the rivers that drove the rise and fall of ancient civilizations throughout the Quaternary[16]. Furthermore, in the Holocene, these floods have contributed to human migration and settlements, river valley civilizations, and their collapse[16].

The geologic perspective begins to highlight the importance of setting a timescale and location of relevance when defining normal conditions. The location of relevance defines the bounds of what area should be considered or investigated. By setting this relevance parameter, one can have a consistent application area, as identifying a hydrological state as a "flood" in the Sahara Desert will be vastly different from that in the tropical rainforests of the Amazon. While this may seem trivial, the previous examples from the geologic perspective highlighted that catastrophic events have shaped the landscape, bringing into question whether information on locations before such catastrophic events can still be considered relevant for identifying floods after such events. The same region or location can then "look" very different if longer or different timescales are utilized, highlighting the importance of also setting the relevant timescale of inquiry. The timescale, as a relevance parameter, sets the period of hydroclimatic history that should be included when calculating exceedance probabilities for the identified region. Similar to setting the rarity threshold, there is no well-defined timescale which should be used; thus, it is up to the individual to identify what duration of information should be included. However, making this decision is nontrivial and has a genuine impact, as the data included in calculating the marginal exceedance probability is based on the identification of the relevant timescale of inquiry. The marginal exceedance probability is therefore influenced by the individual's identification of "what should be the case" in setting what timescale to adopt. In sum, the identification of relevance parameters, the location and timescale of inquiry, as well as the rarity threshold, are normatively influenced.

## How human experiences shape the meaning of floods
Aside from these physical perspectives on floods, there are a number of social dimensions which are relevant for conceptualizing what constitutes a flood. Socio-hydrology is a thriving field with a body of literature which discusses these points in great detail[17-19], while also recognizing the coupling and feedback between social and physical processes[20,21], and the downstream impacts[22].

Historically, floods have greatly influenced the trajectory of human societies, contributing directly to both their emergence and decline[23,24]. Prominent ancient civilizations emerged along fertile floodplains, including in Egypt along the Nile River[25], the successive Sumerian, Akkadian, Babylonian, and Assyrian cultures developed along the Euphrates and Tigris[16], and the Indus Valley Civilization along the Indus River[26,27]. Geological and archeological evidence indicate that floods profoundly shaped societal resilience[16], agricultural practices[28], and settlement patterns[16]. For instance, Nile floods enriched soils, enabling agriculture and the growth of early complex societies[16], yet extreme flooding episodes also forced populations into prolonged migrations lasting centuries to millennia[29]. Upon their return, these groups established foundations for civilization and sustained

settlement[16]. In some instances, catastrophic flooding led to sudden urban collapse, exemplified by the rapid submergence of cities during the 525 AD Euphrates flood event[23]. Conversely, in the Indus region, many urban settlements developed along abandoned palaeochannels, such as the former course of the Sutlej River, abandoned shortly after ~8 ka, where flow had ceased long before Indus occupation (~4.6–3.9 ka). These relict valleys provided stable, well-drained ground and accessible groundwater, rather than exposure to active flooding[26]. Together, this illustrates the dual role floods have played in human history, both fostering and disrupting human settlements and profoundly shaping patterns of habitation throughout history.

An additional aspect of normativity is then present within the conceptualization of "flood", where a specific flood can be considered good or bad based on the conception of desirability that an individual adopts. For example, if an individual thinks that the spread of sediments and the subsequent renewal of the soil is desirable, they may consider the flood to be good. Conversely, an individual living in that region who was forced to move may have a different notion of what is desirable and view the flood to be bad. Ultimately, one's notion of desirability regarding the presence of the water will greatly influence their perception of the flood phenomenon.

Adopting a notion of desirability can also be seen in the socioeconomic perspective on floods, which focuses on risk metrics, specifically physical risk. Generally, these metrics are based on three components: the hazard, the exposure, and the vulnerability related to flood events. A hazard is conceptualized as the likelihood of such an event occurring, where the quantified probabilities of different return periods are used to represent the likelihood of a given event. While the relevance parameters and the return period itself are normatively influenced, the assessment of exposure and vulnerability requires further normativity. The identification of the aspects of society which are included, or should retain the focus, rely on an additional human conception of what is important or desirable to protect.

Historically, within the hydroclimatology literature, studies on the exposure for flooding have focused on losses related to property damage, macroeconomic indicators, and household-level losses[30]. In practice, monetary loss and the loss of human life are generally considered to be potential impacts of flooding and retain the focus[30]; however, other potential impacts, such as cultural loss, or additional aspects of impact, such as the temporal duration, are often not present in these discussions. In recent years, there have been efforts to highlight or discuss additional aspects of exposure, such as secondary impacts that arise due to an initial impact, when discussing flood risk[31]. However, when constructing our risk metrics, there is no objective means of identifying what is or is not included in exposure, or what is a relevant impact to consider. What we consider to be impactful is then heavily normative, relying on a human conception of desirability to identify what impacts should be considered.

The use of normativity also extends to how vulnerability is conceptualized. Vulnerability is generally defined as the degree to which a system is or is not able to cope with some adverse pressure or stressor[6,32]. Within the flood risk context, vulnerability has been described as "the most important component of disaster mitigation"[33]. While physical and social vulnerability are prominent conceptualizations[33–35], there is not a preferred conceptualization of vulnerability utilized across studies. When designing specific methods for weighing vulnerability, one must consider the major dimensions of these conceptions of vulnerability and reason as to how to bring together potentially vastly different aspects of vulnerability into a metric. However, there is no single way to do this. In practice, individuals must identify the specific aspects of the system which they believe should be included and how these aspects should then be weighed against other relevant aspects. How vulnerability is identified and utilized is then reliant on a notion of desirability.

The choice of which impacts retain the focus and the methods of weighing vulnerability then have a subsequent effect on the interpretation of the apparent risk of flooding events. For example, focusing on macroeconomic indicators rather than on welfare indicators[30]. If one conceptualizes risk based on macroeconomic impacts, there is the potential that

individuals who have greater wealth could be seen as being at greater risk, as they, under this perspective, have more to lose. Conversely, if welfare is taken as an indicator, than already disadvantaged individuals or communities could be seen at greater risk due to their already marginalized position within society.

Regardless of how exposure and vulnerability are specifically identified in practice, one point remains salient. In discussions of flood risk, the use of conceptions of desirability have resulted in floods being exclusively seen as a hazard, where flooding is something that must be managed. This has affected our view of floods, resulting in a negative framing to often be associated with the phenomenon.

## A flood demarcation framework

The physics-based and social perspectives on floods have highlighted key features of the phenomenon, where the use of normativity is a necessary aspect of identifying this phenomenon. However, most current working definitions neither explicitly reference nor fully capture the extent of the interactions between the normative and physical aspects, making them inadequate. A reconceptualization of the term, which both recognizes and explicitly references these essential aspects, is then necessary. While such a reconceptualization can begin with transient bodies of water as objective empirical entities, a discussion of when such bodies are considered floods will rely on some subjectively-identified conception of desirability, or, what one may call, a meaningfully bounded perspectivism. Such a conceptual space can be seen in Fig. 1.

Within the hydrologic and flooding context, any definition of "flood" must include a discussion of water. In the most general sense, this would be the pooling of water and movement of water, or fluxes. The pooling of water refers to locations where water is present, such as in lakes or glaciers. Fluxes of water refer to when and how the water moves, such as during evaporation, precipitation, and streamflow. These two aspects highlight the key physical variables of the concept, (1) the location of water, and (2) the movement or potential for the amount of water to change.

However, this discussion of pools and fluxes is far removed from any meaningful discussion of a flood, as it misses one key aspect. As demonstrated in Section "Current Perspectives on Floods", any conception of a flood has some degree of normative influence. This provides the first demarcation within our conceptual space. The normative influences on the objective aspects of pools and fluxes of water orients the discussion away from pools/fluxes to focus on what we call physical floods.

### Physical floods

Normativity, some human conception of desirability or identification of "what should be the case", identifies the bounds of the two relevance parameters of investigation, the timeline and the location, as well as the rarity threshold. Once these parameters and the rarity threshold have been set, it is possible to identify a physical flood. Thus, physical floods constitute pools or fluxes at various exceedance probabilities, where the relevance parameters, the timescale and location, and rarity threshold are normatively identified.

From this, the practical identification of physical floods can be done. The timeline and geographic region of interest provide the bounds of what should be considered normal conditions. These parameters can be seen as providing relevance, defining what data is included in the inquiry. However, as highlighted in Section "How Physical Factors Contribute to the Conceptualization of Floods", the selection of the rarity threshold is also dependent on a notion of "what should be the case". Specifically, one must identify what rarity threshold should be considered far enough on the tail of the distribution to constitute a physical flood. As we have discussed, rarity thresholds will often be identified by return periods, in line with the physics-based perspectives. However, this need not be the case; rarity thresholds can also be identified with a Bayesian conception of rarity (see Section "Potential Challenges and Outlook").

While setting the rarity threshold will tend to be done on human scales, such as a few years to 100s of years, this is merely due to the adoption of a

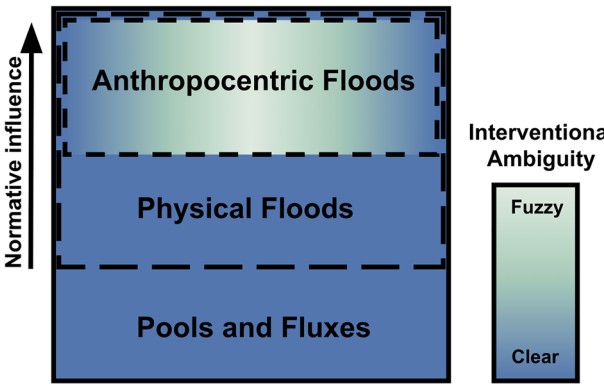

**Fig. 1 | A didactic schematic for our demarcation framework.** The introduction of normativity, primarily in identifying a rarity threshold, demarcates physical floods from ordinary pools and fluxes. As normative influence increases, we move from physical floods to anthropocentric floods, which utilize notions of desirability for their identification. Differences in these notions among individuals result in interventional ambiguity (see Section " Interventional Ambiguity"), where in some cases it is clear that the anthropocentric flood should be intervened on and in other cases the need for intervention is unclear, or fuzzy.

threshold that is deemed relevant for humans. However, this is not necessitated by any physical principle. Take the following thought experiment. Say there is a species of humans who live vastly longer lives, on the scale of 1000s of years. Their conception of a desirable threshold to adopt could, in turn, be much longer. They would then utilize their own desired threshold to investigate if a region is experiencing an amount of water which would be beyond normal. From this, what we may see as a catastrophic riverine flood could be, based on their threshold, a relatively periodic—and thus expected —ebb and flow of the fluvial system. Under this perspective, some things which may have previously been considered a flood could, under a differently defined threshold, not be considered a physical flood. The set of pools and fluxes which are identified as physical floods can then vary based on this normative influence, signified by the dashed line demarcating the set from pools and fluxes in Fig. 1.

This way of identifying a physical flood and the direct reference to the normative influence has a subsequent impact on our conceptions of second-order floods. Take the case of coastal floods as an example of a second-order physical flood. To identify these floods, the location of investigation would be constrained, in part, due to this "coastal" aspect of the second-order flood. However, the timescale would not be constrained based on the flood's second-order nature. Identifying the timescale would still be based on a notion of desirability, which identifies what timescale should be utilized.

Now, it is well established that anthropogenic climate change has increased the prevalence of water in coastal areas[2], with a higher likelihood of many coastal areas being submerged than in the past. With this shift, the extremes are now becoming the new normal. How one sets their timescale of inquiry then has a substantial impact on the identification of coastal floods. One could set their timescale to be only the recent past, when anthropogenic climate change had already caused coastal areas to have a higher likelihood of being submerged. If this is the case, then such an area, which used to experience periodic coastal flooding, would stop being considered flood-prone. Rather, this area would begin to experience more consistent pools and fluxes of water as examples of inundation[36], where the presence of the water is no longer sufficiently at the tail of the distribution to be considered a physical flood. Thus, an argument can be made that such an area has simply become a part of the sea or ocean.

Recognizing this reliance on the normativity involved in setting these relevance parameters is additionally impactful. Through this recognition, one can begin to properly situate the identification of second-order floods in the relevant context. Just as in the case of coastal floods, the recognition that the hitherto extremes are becoming more frequent provides additional justification to curb carbon emissions, as areas are now entering into this

new state of "normal". Additionally, the recognition of the importance of these parameters provides the language to better identify or characterize our research questions. Here, one can place their questions in the correct context and clearly identify what set of parameters they are adopting and articulate the reasoning behind why those specific parameters were used.

## Anthropocentric floods
While the phrase anthropocentric floods (not to be confused with anthropogenic floods) has made its way into the scientific literature[37], it has not been systematically defined. We argue that the introduction of an additional aspect of normativity can, again, provide a demarcation within the conceptual space and identify the set of anthropocentric floods as a subset of physical floods (as shown in Fig. 1).

The additional aspect of normativity is specifically regarding adopting a notion of desirability for the presence of water at the identified time and location. This notion of desirability would, generally, be provided by an individual's personal values. The use of values in science has been widely discussed within the philosophy of science literature[5], where these personal values assist in the production of an individual's judgment[38] and contain the normative content necessary to identify if the presence of water is desirable or undesirable[39]. Take the following example: one may use the social value of economic prosperity to argue that a physical flood, which resulted in damage to a city and further monetary loss, is undesirable. The identification of this physical flood as being undesirable would, in turn, make this specific event an anthropocentric flood. The normative evaluation of the absence/ presence of water is then the necessary condition for moving the discussion from one of physical floods to anthropocentric floods.

Physical floods are then anthropocentric floods only when a notion of desirability is used to claim that the water should be there or should not be there. A flood on some distant planet or in the distant geological past can be identified as a physical flood if one normatively defines the relevance parameters and rarity threshold; however, if a notion of desirability for the presence of that water is not used, then this would not be considered an anthropocentric flood. However, the set of physical floods that are identified as desirable/undesirable will vary depending on one's values, resulting in a corresponding variability in the set of physical floods that are also identified as anthropocentric (signified by the dashed line demarcating this set in Fig. 1).

The influence of adopting different values on the conceptual space can be seen in the following example of three individuals. When thinking about floods, the first individual adopts techno-optimistic values, the second adopts naturalistic values, and the third is generally agnostic and doesn't have strong views on floods. For each of these individuals, their set of physical floods can be the same if they set the same relevance parameters and adopt the same rarity threshold. However, their sets of anthropocentric floods will differ greatly. The first individual may hold the view that all built infrastructure should be protected, and any physical flood that has the potential to impact this infrastructure is undesirable. In turn, their set of anthropocentric floods will be bounded accordingly. Alternatively, the second individual may hold the view that many floods are beneficial to ecosystems, providing nutrient-rich sediments, where their set of anthropocentric floods is defined by the conception of desirability this value provides. These two individuals' sets of anthropocentric floods may have some overlap, where certain instances are shared by both individuals, while each may also exclude instances that the other has included.

While these two individuals may not agree on the (un)desirability of any specific instance of a physical flood, due to their adoption and utilization of values, their sets of physical floods, which are identified as being anthropocentric, will have relatively similar sizes. This puts these two individuals in contrast with the third individual. The third individual will generally be agnostic to many physical floods, not deeming them desirable nor undesirable unless the flood impacts them personally. The third individual's set of anthropocentric floods will then be smaller in magnitude, taking up less space within the set of physical floods, when compared to the other two individuals. The set of anthropocentric floods can then vary

greatly dependent on what values one adopts or the lack of utilizing values in general.

On the surface, this may seem to be a negative consequence of the demarcation framework. However, we see this variability as a necessary feature of any usable definition of the flood phenomenon. Specifically, by relying on normatively-defined relevance parameters and rarity threshold as well as the adoption of a conception of desirability for the presence of water, any definition of flood must allow for such conclusions to be drawn. While what is included in the sets of floods may vary between individuals, the introduction of a normative influence is what makes a flood a "flood". Our demarcation framework recognizes that floods are a fundamentally human-oriented phenomenon, where even the relevant features of the physical phenomena cannot be separated from the normativity used to identify them.

## Interventional ambiguity

Through the adoption of a notion of desirability, some anthropocentric floods are viewed as a hazard, where the presence of the water is viewed as undesirable, and some as a benefit, where the presence of water is viewed as desirable. However, this reliance on a notion of desirability leads the demarcation framework to maintain varying degrees of what we call interventional ambiguity.

The identification of what is desirable is always done in a decision-making context where a level of normative uncertainty is present. This form of uncertainty is distinct from empirical uncertainty, what has been described in the engineering literature as ontological uncertainty[40], which is the uncertainty regarding "what is the case"[41]. Normative uncertainty, rather, is regarding "what should be the case"[41]. The presence of this type of uncertainty is due to the fact that one must identify whether the presence of the water is desirable or not.

While an individual's personal values can be used to identify a notion of desirability necessary to make such a decision[39], there is no preferred notion of desirability outside the individual person. Consequently, adopting a notion of desirability can be seen as subjective, or dependent on some aspect of the individual person[39]. Within these decision-making contexts, different values can be used to evaluate the same event, leading to different conclusions. Take the following example. A specific physical flood event can be evaluated using the social value of ecological health, leading one to see the restoration of nutrients to floodplains as a benefit, or the social value of human health, leading one to see the regional displacement and potential increase in waterborne disease as a hazard. According to one value, the flood is beneficial, and according to the other, it is a hazard. However, it is unclear which of these values to prefer and utilize. The presence of this uncertainty has genuine implications for our decision-making, as whether to do something, i.e., managing or intervening on a given event, and subsequently what to do, is dependent on whether the event is viewed as a hazard or benefit. Thus, interventional ambiguity is an emergent feature of this conceptual space, which recognizes the uncertainty present within the context of identifying and adopting notions of desirability given their use in decision-making on what should be done in any specific instance of an anthropocentric flood.

Interventional ambiguity can be placed on a qualitative scale, from clear to fuzzy, representing the varying degrees of this ambiguity across the conceptual space (as depicted in Fig. 1). The largest area of the conceptual space, which has interventional clarity, is outside the set of anthropocentric floods. Specifically, the regions of pools/fluxes and physical floods have no interventional ambiguity, where the lack of interventional ambiguity for these cases is necessary by definition. Pools/fluxes and physical floods do not rely on identifying if they are desirable and then cannot be seen as good or bad; in turn, we need not question if they should be intervened on. Anthropocentric floods are then the only region of the conceptual space where a notion of desirability is used to identify if the presence of water is good/bad. For anthropocentric floods, there are two instances of interventional clarity, where the ambiguity is negligible or approaching zero, (1) cases where it is clear that intervention is necessary, and (2) cases where it is clear that intervention is not necessary.

To achieve interventional clarity for anthropocentric floods, any single individual can consider a situation to be interventionally clear if they have a well-defined set of values that can provide the normative content necessary to address the uncertainty and adopt a notion of desirability. However, decisions to intervene generally rely on a larger set of individuals reaching an agreement on desirability. In such situations, the focus rests on the aggregation of judgments, where the views of many individuals are combined using a defined method to produce a single judgment[42]. There are some examples, such as the protection of human life, which are widespread across societies and can, in some sense, be seen as widely agreed upon. Instances of tsunami inundation may then be argued as such an example of an anthropocentric flood, which is universally undesirable. While the widespread agreement on the undesirability of tsunami inundation may be true, relative to those individuals experiencing the inundation caused by the tsunami, such agreement on the adoption and utilization of the value of protecting human life cannot be universally guaranteed. Take, for example, the case of COVID-19, where other values, such as monetary loss, conflicted with the values for the protection of life and public health[43]. Thus, even for a well-established value, which may on the surface appear to be universal, the utilization of the value in decision-making is not guaranteed. Ultimately, there are various existing problems associated with synthesizing or aggregating perspectives on desirability[42], where these problems can begin to highlight ambiguity regarding if something should be done for a specific anthropocentric flood.

When agreement cannot be reached, meaning there is not interventional clarity, we find ourselves in the much larger area of the set of anthropocentric floods, those fuzzy cases where it is unclear if there should be intervention. These degrees of interventional fuzziness can be seen in a variety of examples, such as when individuals adopt different values, when shared values lead to divergent conclusions, when different preference orderings of shared values are used, as well as others. These cases of interventional fuzziness can essentially be seen as a product of some form of disagreement on the values used when identifying and utilizing a conception of desirability.

However, by recognizing the presence of interventional ambiguity, we can provide a better understanding of when intervention should take place and also why, in some cases, intervention is not occurring. Specifically, this ambiguity provides insight into the problem of the willingness to act. Here, the incongruence in social values can lead to disagreement, where it may be unclear if an anthropogenic flood should be intervened on. These disagreements in values can be due to the political and social environment one finds themselves in, as an individual's values are often shaped by social pressures. Thus, if there are incongruous values or differences in values, caused by social or political differences, there may be a subsequent impact on an individual's priorities or views on what should be done[44]. This can result in limits to preventing specific anthropocentric floods that some people, but not enough to reach agreement and induce action, have deemed to be undesirable.

Nevertheless, even when there is agreement on the undesirability of a given anthropocentric flood, action cannot be guaranteed. When an anthropocentric flood is deemed undesirable by a group, the use of values was such to reach agreement on the flood's undesirability. While this guarantees that action on management/prevention is deemed good, or a benefit, it doesn't guarantee that decision-makers will take that action. This can be due to potential differences in preference orderings, where there is disagreement on how large of a problem the anthropocentric flood is relative to other issues. While all the decision-makers may view the flood as a problem worth addressing, there may not be agreement among the group as to if this is the top priority. This presents a problem with the willingness to allocate potentially limited resources to managing the anthropocentric flood over competing interests or other problems deemed to be of greater significance. In this way, by recognizing the interventional ambiguity present in the set of anthropocentric floods and the additional normative influences, one can better identify these limits to the management/prevention of floods.

**Perspective**

## Potential challenges and outlook

Specifically, by explicitly recognizing the use of notions of desirability, there is more conceptual leeway to avoid past mistakes of overemphasizing engineered "solutions" that are later recognized as harmful. For example, in the last century, the straightening of many river channels, in the service of flood management, gave way to large-scale river restoration projects after harms were recognized[45]. Our proposed demarcation framework hopes to foster critical questions around the conceptual inertia in flood-risk management before these problems arise. Within our framework, one can now ask whether what we are managing are even physical floods to begin with, as if we are to set a different rarity threshold, we may rather be dealing with events that should be characterized as acceptable distributions of pools/fluxes, not being considered physical floods at all. Once we use this broader conceptual paradigm, that avoids calling any and all confrontation with water a de facto "flood", we can have richer discussions around facilitated relocation from waterfront properties that all too often face inundation[36]. This could also allow for additional approaches, such as anthropocentric approaches[46], to be undertaken for flood risk management, where a recognition of the complexity of the situation can be beneficial[47].

Our proposed demarcation framework is not without its potential challenges, however. Some may argue that the many catastrophic palaeo-floods on Earth and the megafloods inferred from other planetary surfaces, such as Mars[8,9,12,14], as well as other man-made dam failures[13] (which are not well-represented as samples from a probability distribution representing relative frequencies[9,10,12]), cannot be captured within our framework. However, we contend that such events can still be considered rare and unexpected within the scope of a Bayesian interpretation of probabilities. Within our framework, we do not specify how one should identify or even conceptualize the rarity threshold they adopt. If one was to adopt a Bayesian interpretation of rarity, where rarity is then a statement about one's beliefs, such events can still be considered rare. Such a rarity threshold would then be referencing a low subjective probability given what the individual eliciting the probability knows and not a low frequency in some mind-independent sense.

A second potential challenge related to the rarity threshold could stem from concerns about regular or seasonal flood events. In many environmental settings, annual inundation is predictable and part of the normal regime rather than exceptional. With this in mind, some may argue that rarity is not applicable in these contexts. While for the vast majority of cases the normatively-defined rarity threshold of physical floods can be understood in terms of exceedance probabilities of stochastic events, the same concept also applies to settings where pools and fluxes change predictably and periodically. In such cases, for example, the annual flooding of the River Nile can be considered a physical flood because of its temporal rarity within a year. This further highlights the importance of how the normatively-defined relevance parameters and the rarity threshold are set. Our framework can, indeed, accommodate this type of periodic flooding by explicitly specifying whether the relevant event is a deviation from a seasonal baseline, exceedance of a management threshold, or a substantial change from historical distributions over specified timescales.

Moving forward, our framework provides conceptual clarity on the normativity involved in defining "flood", thus aiding discussions on climate hazard management. By explicating the normative aspects, researchers can produce work that is not only scientifically rigorous but also more transparent about the adoption of conceptions of desirability through the use of values[48]. The transparency can provide clarity on which values are used by the researcher and, in turn, which values are embedded in the research[49]. This transparency can assist in reducing the opacity of many practical decisions which must be made in hydroclimatic contexts. Further, it may enhance legitimacy by allowing stakeholders to evaluate whether those values align with community priorities and allow them to function as objects of democratic deliberation[50,51].

In this paper, we posed a question to the hydroclimatology community —what is a flood? Through this endeavor, we hope to foster new dialogue within the community on how to best conceptualize "flood", which could lead to the refinement of, or conceptually fruitful challenges to, our proposed demarcation framework.

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

## Acknowledgements

The authors would like to thank Berina Kilicarslan, as well as the three reviewers, for their helpful comments. Mason Majszak would like to thank the Swiss National Science Foundation for financial support (grant P500PH_225416).

## Author contributions

M.M., O.W. conceptualized the project. O.W. supervised the project. M.M. wrote the initial manuscript, with contributions from A.Z., O.W. All authors commented on and revised the manuscript during the writing phase and reviewed the final version. All authors helped with addressing reviewers' comments.

## Competing interests

The authors declare no competing interests.

## Additional information

**Supplementary information** The online version contains Supplementary material available at https://doi.org/10.1038/s43247-026-03491-2.

