## [Transparent Peer Review File · Communications Earth & Environment]

Defining flood

Corresponding Author: Dr Mason Majszak

Version 0:

Decision Letter:

Dear Dr Majszak,

Your manuscript titled "What is a flood?" has now been seen by 3 reviewers, and we include their comments at the end of this message. They find your work of interest, but some important points are raised. We are interested in the possibility of publishing your study in Communications Earth & Environment, but would like to consider your responses to these concerns and assess a revised manuscript before we make a final decision on publication. A revised version must clarify the scope of applicability of the proposed norms and expand the discussion to more broadly address the role of values beyond normativity.

We therefore invite you to revise and resubmit your manuscript, along with a point-by-point response that takes into account the points raised. Please highlight all changes in the manuscript text file.

Please submit your point-by-point responses as a separate file, distinct from your cover letter where you can add responses to the Editors' comments that you do not want to be made available to the reviewers. Word files are preferred. We recommend that any figures, tables or graphs that are included in the response to reviewers are also included in the main article or Supplementary Information.

Please use the following link to submit your revised manuscript, point-by-point response to the referees' comments (which should be in a separate document to any cover letter), a tracked-changes version of the manuscript (as a PDF file) and the completed checklist:

Link Redacted

We hope to receive your revised paper within six weeks; please let us know if you aren't able to submit it within this time so that we can discuss how best to proceed. If we don't hear from you, and the revision process takes significantly longer, we may close your file. In this event, we will still be happy to reconsider your paper at a later date, as long as nothing similar has been accepted for publication at Communications Earth & Environment or published elsewhere in the meantime.

Please do not hesitate to contact us if you have any questions or would like to discuss these revisions further. We look forward to seeing the revised manuscript and thank you for the opportunity to review your work.

Best regards,

C. Kendra Gotangco Gonzales, PhD
Editorial Board Member
Communications Earth & Environment
orcid.org/0000-0002-3436-9813

Yann Benetreau, PhD
Consulting Editor, Communications Earth & Environment
Deputy Editor, Communications Sustainability
Nature Portfolio
NY office

EDITORIAL POLICIES AND FORMATTING

- Behavioural and social science
- Ecological, evolutionary & environmental sciences
- Life sciences

Furthermore, please align your manuscript with our format requirements, which are summarized on the following checklist: <https://www.nature.com/documents/commsj-phys-style-formatting-checklist-article.pdf> Communications Earth & Environment formatting checklist

and also in our style and formatting guide <https://www.nature.com/documents/commsj-phys-style-formatting-guide-accept.pdf> Communications Earth & Environment formatting guide .

*** DATA: Communications Earth & Environment endorses the principles of the Enabling FAIR data project (<http://www.copdess.org/enabling-fair-data-project/>). We ask authors to make the data that support their conclusions available in permanent, publically accessible data repositories. (Please contact the editor if you are unable to make your data available).

All Communications Earth & Environment manuscripts must include a section titled "Data Availability" at the end of the Methods section or main text (if no Methods). More information on this policy, is available at <http://www.nature.com/authors/policies/data/data-availability-statements-data-citations.pdf>.

If a community resource is unavailable, data can be submitted to generalist repositories such as <https://figshare.com/> or <http://datadryad.org/> Dryad Digital Repository. Please provide a unique identifier for the data (for example a DOI or a permanent URL) in the data availability statement, if possible. If the repository does not provide identifiers, we encourage authors to supply the search terms that will return the data. For data that have been obtained from publically available sources, please provide a URL and the specific data product name in the data availability statement. Data with a DOI should be further cited in the methods reference section.

REVIEWER COMMENTS:

Reviewer #1 (Remarks to the Author):

The subject of the Perspective by Majszak et al. and the concepts discussed are important and timely. The question, "What is a flood", is thought-provoking, because I think that many of us, as the authors suggest, just take a flood to be an objectively determined physical event. Their framing of floods as having a normative component relating to what individuals subjectively

think about their situation (e.g., location and timeframe) seems sound to me based on my work. The authors might find the following to be relevant to their research, especially regarding timeframes and risk tolerance/aversion:

• Lutz, T., 2024, A Review of the Philosophy of Flood Risk Communication and Education: New Insights Informed by Critical Complexity: *Water* 16 (23) 3459, <https://doi.org/10.3390/w16233459>

The Perspective is organized straightforwardly to bring out the idea that a sound conception of floods requires a normative component: Section 2 (Perspective on Floods), Section 3 (... Socio-physical conceptualization...), and Section 4 (Interventional Ambiguity) form the heart of the paper. "Ambiguity" is key because failures in flood management are often seen as technical problems, to be addressed by better engineering or better communication of status quo ideas (e.g., Q100). The authors recognize that in complex systems (e.g., societies) there is always an incompleteness or ambiguity that needs to be considered.

Despite all the good things to be said about the paper, I think substantial rewriting should be required before publication occurs. I was driven to distraction by several things.

- Inefficient writing slows down and burdens the reader. More than 1 of every 7 lines contains a sentence beginning with an unnecessary or vague word or phrase such as "This" (26x), "Here" (17x), "Thus" (7x), "In turn" (5x), "In this way" (29x). Usually such wasted words can be eliminated by simply writing clearly in the first place so that it doesn't become the reader's job to figure out what the author is trying to say.
- The paper's subject requires the words "normative" and "should" to be used a bit. But the fact that they appear 105 times in the text (once in every 5 lines, on average) is mind-numbing for the reader. It makes one wonder what's really new and what's just repetitive. I think the authors may be trying to make sure the reader is catching on to the novelty of their ideas but they are overdoing it! The paper can be written less repetitively.

The figures have their own issues.

- Captions for Figures 2 & 3 go beyond describing the figure to include explanation that should be in the text.
- I question whether Figure 1, which is complicated, contributes essential information. I think the importance of timeframes and locations is adequately communicated in the text; this figure could be eliminated without affecting the impact of the paper.
- Figure 2 is inconsistent and confusing. If normative influence increases upward (arrow on left) why do the 'Physical' and 'Pools and Fluxes' boxes extend above the Anthropocentric box? Why is there a gradational Clear-Fuzzy bar on the right when there is essentially no gradation shown in the figure? And, why is this concept illustrated as 2-D? The horizontal direction has no axis and seems to be there only to turn a 1-D concept into 2-D diagram.
- Figure 3 is inconsistent and confusing, too. Shouldn't each of the three sub-diagrams have their own set of axes? Or, why do the A.F. boxes inside have different sizes and locations within each physical flood box? More substantially, to clarify the text for the reader, I think that the hypothetical values should be replaced by real-world examples. Otherwise, I don't think it's clear what the "values" might be. Since the attributes of the three individuals are given in the text (lines 390-396), why not make the diagram more explicit?

To recap: This paper brings forth interesting and important insights into a "simple" question: What is a flood? Many in the readership may be led to expand their thinking about flood risk and flood communication, especially with regard to interventional ambiguity. However, whether readers are able to get the intended ideas depends on improving the writing and figures so that the message gets across.

Reviewer #2 (Remarks to the Author):

Review of "What is a flood? Reviewed by Victor R. Baker, Dept. of Hydrology and Water Resources, The University of Arizona, Tucson, AZ. 85721 baker@arizona.edu

GENERAL COMMENTS

This is a philosophical paper in the tradition of "providing lawyer-like answers to child-like questions." Philosophy can raise questions about fundamental concepts that may be assumed without considering the consequences of their application. The manuscript under review addresses concerns about the concept "flood" in diverse applications of that concept. It advocates a couple of demarcations, that by definition fix the boundaries or limits on what it is to be a flood.

There is an important and relevant motivation for this project, as follows, "...our proposed demarcation hopes to foster critical questions around the conceptual inertia in flood-risk management before...problems arise..." (lines 529-531). By developing what is proposed to be a demarcation between physical and anthropocentric floods, "...one can now ask whether what we are managing are even physical floods to begin with..." (lines 531-532). What the authors term "this broader conceptual paradigm" will avoid, "...calling all confrontations with water a de facto flood," and "allow for additional approaches, such as anthropocentric approaches...to be undertaken for flood risk management..." (lines 535-539).

The problem to be solved by this demarcation project is termed "interventional ambiguity," which is, "...the lack of consensus in communities...on whether to seek engineered solutions to specific flood-related problems..." (lines 27-29) The manuscript does not claim to resolve this issue. Instead, it hopes that the perspective that it provides, "...fosters dialogue within the hydroclimatology community on how best to conceptualize this phenomenon, which could lead to further refinement of this proposed demarcation framework" (lines 559-562).

The proposed “conceptual paradigm” to deal with the problem of “interventional ambiguity” is the recognition of a need for “...conceptual clarity...[in the] use of the term and the impacts of the events in question...” (lines 52-54). The problem of the term “flood” referring to different “conceptions of the phenomenon” is to be resolved by incorporating “normativity” (what ought to be) in regard to both the physical and the anthropocentric answers to question about the meaning of word “flood.” Physical floods are defined as, “...pools and fluxes beyond a normatively-specified rarity threshold – the exceedance probability” (lines 22-23). “Anthropocentric floods are then physical floods deemed desirable or undesirable” (Line 25).

Normativity determines what ought to be in regard to a norm, that is in regard to some standard. For the physical floods this standard is determined by “a mathematical definition,” the exceedance probability, which given by an equation. While this is claimed to be a physics-based standard, it seems to be more to be an engineering criterion used for evaluate risk, which would be the multiplication of this probability standard times a consequence.

One problem here is that many physical floods do not have probabilities because they are singular events. Constructed dam failures occur because of design defects, or even human negligence. Extreme high-energy floods may occur because of unique circumstances that have nothing to do with a time sequence of river discharges, storm events, or other phenomena subject to estimates of their likelihoods. The proposed norm does seem applicable to the hydroclimatological riverine flooding phenomena (a subset of “what is a flood?”) that is the apparent focus of this manuscript.

For the anthropocentric floods the normative standard is to be a value judgment as to whether a flood is desirable or undesirable. This achieves a demarcation, which is a fixing of the boundaries or limits on something, in this case, the concept of a “flood.” As noted above, this value judgment is directed making management decisions “before...problems arise.” This notion of making a prior judgment is also an engineering-oriented criterion that needs to be distinguished from a pragmatic/scientific approach that will be noted below.

DETAILED COMMENTS

The Very Limited Conception of “Flood”

The manuscript should not imply that it is answering a very general question concerning the concept “flood.” It is actually dealing with very limited aspects of that phenomenon. It does not deal at all with the reality that ancient massive floods have characterized both Earth and Mars (Baker and Milton, 1974; Burr et al., 2009; Baker, 2020).

It also does not deal with man-made (i.e., “human” floods that would involve a different wrinkle on what is subject to the demarcation: “anthropocentric floods.” Famous examples are the many dam failure floods that involve human negligence. There is a rather large literature on these (multiple books) with detailed case studies, There are also examples of massive flooding being used as a weapon of war, as occurred in northern China during the 1930 (leading to massive losses of life and the displacement of millions of people). Flooding caused by malicious human intent includes past Mississippi River levee ruptures in Black-populated parishes upstream of New Orleans that were purposely perpetrated to reduce river flood stages that might otherwise damage white-owned business interests in that city.

The “physical floods” demarcation seems to ignore seasonal inundations, which occur every year. The concept of exceedance probability is irrelevant to such “floods.”

There clearly are important human aspects of the risks posed by tsunami floods, but these are not susceptible to being considered “desirable” relative to human interests. There is no ambiguity whether such floods are desirable or undesirable; they are always undesirable in regard to humans that are placed at risk.

Indeed the emphasis on exceedance probabilities and other comments in the text indicate that the physical flooding focus of this manuscript is on engineering issues related to hydroclimatic riverine flooding, rather than on scientific concerns that might otherwise be suggested by the title of the manuscript. More will be said on this below.

Geological Perspective – lines 112-135

This section is limited in its demarcation of what it means to be “geological.” As a geologist, I find the whole section to be so misleading that it might be best to just eliminate all of it, including Figure 1.

This section of the text really does not add to the main objective of the paper, which, as noted above, concerns hydroclimatic riverine floods, not floods in general. Eliminating this section would shorten the manuscript, and it would also focus it more on the what seems to be the main objective.

For some a more general view on how geological thinking can relate to flooding, including its “anthropocentric” aspects, one can see Baker (1994).

Though eliminating the section is the easiest path to revision, acknowledging the limited aspect of its whole approach – that it considers concept “flood” largely from an engineering viewpoint – would be helpful to the reader.

Figure 1 - Distribution of archaic humans, prehistoric sites, and ancient civilizations on active global floodplains and flood-related landforms.

The base map for this figure (Panel A) is inappropriate. It emphasizes global relief, much of which is submarine - having nothing to do with the “flooding” that is of concern in the manuscript. Most of the elevation scale scale is devoted to submarine relief (0 to -11 km depth).

The legend refers of “prehistoric outburst floods” and supposedly shows locations with blue dots. However, there is only one blue dot that I can see on the map, and it is on the Arabian Peninsula where I know of no outburst flood evidence. Actually there are scores of ancient outburst floods that have been documented and mapped on multiple land areas, mainly near the margins of the large Pleistocene ice sheets that formerly covered parts of Eurasia, North and South America (e.g., Baker, 2020; Baker and Carling, 2022).

The locations of “Holocene Human Floods” are to be indicated as large purple dots (from the caption), but they seem to be shown as small reddish dots on the map. Also, the concept of “Holocene Human Floods” is not defined in the text, which focuses on “anthropocentric floods” defined in terms of their desirability or lack thereof. During the Holocene there were humans all over the land areas of the map (except Antarctica).

The distribution of “Archaic humans” is to be designated by small brown dots according to the caption, but the assumed distribution is shown by large brown dots on the map. The legend for “human presence” defines “Archaic” as between 2 Ma and 30 ka. However, humans are now known to have arrived in Australia by 40-50 ka. There is also mounting evidence suggesting humans in the Americas prior to 30 ka, and humans have long been in southern Africa.

Finally the distribution of “global floodplains” is certainly not shown globally. There are many floodplains in northern North America and northern Eurasia. The distribution seems to be arbitrarily cut off at a latitude of 60 degrees north. Many of what are mapped as “floodplains” are actually be seasonal wetlands. Moreover, the high density of “Holocene floods” obscures the the pattern of floodplains in many of the areas where “anthropocentric floods” would be of most interest, as in western Europe, eastern China, and the eastern U.S.

There are so many problems with this map that it would be simplest to just eliminate it along with the relevant text.

Physical floods – defined as “...pools and fluxes beyond a normatively-specified rarity threshold – the exceedance probability”

“Pools and fluxes” characterizes physical phenomena that are not floods. Oceans are pools of water that have huge fluxes (ocean currents). These pools and fluxes are physical, and they have nothing to do with exceedance probabilities.

There are also floods, known as megafloods, with fluxes that are measured in the same units (sverdrups – millions of cubic meters of water per second) as ocean currents (e.g., Baker and Carling, 2022). There are also cases when oceans (pools) can generate fluxes that produce megafloods over the land. This occurred 5.33 million years ago when the Atlantic Ocean spilled over the divide that previously separated what is now Spain and Morocco. The resulting “Zanclean Flood” created what is now the Strait of Gibraltar as it spilled into the then-dry Mediterranean Basin. What is now the English Channel was created by mega-flooding. This also occurred with the straits connecting what is now the Black Sea to the Mediterranean.

These various megafloods are definitely physical, but the concept of exceedance probability has nothing to do with them. Exceedance probabilities are used in engineering evaluations of risk (consequence time probability) for purposes of design, mitigation works, etc. Extremely large physical floods are essentially unique events for which the concept of probability is irrelevant. There is no quantitative measure of expectation for the likelihood that such a future event will occur because the circumstances that produce these events are unique to the special circumstances of the events themselves. The problem is not analogous to the counting of colored beans in a jar. There is only the certainty that what has happened can indeed happen.

I could go on at great length about ancient floods on the planet Mars and much else, but the main conclusion here is that the whole argument of this manuscript is best limited to hydroclimatic riverine floods.

Anthropocentric floods – defined as “physical floods deemed desirable or undesirable”

The complexity of human decisions that have to be made in regard to flooding are well illustrated by the very recent paper on where formal rule curves intersect with human discretionary judgment in regard to reservoir operations relative to flood risk (Gautam et al., 2025). It would be well to check how the demarcation of “anthropocentric floods” fits with this aspect of flood hydrology.

Normativity

Normativity is only one approach to the general issue of values. An alternative approach would be to consider the complete axiology of the situation (e.g., Baker, 1998).

As noted in the discussion of “interventional ambiguity” the focus of this manuscript is what is desirable or undesirable in regard to the physical phenomenon of (hydroclimatic riverine) flooding. It is claimed (lines 363-364.) that, “...normativity is specifically regarding adopting a notion of desirability for the presence of water at the identified time and location.” In other words, what an individual wants or wishes for (desires) as a course action is to be specified by what ought to be (normativity)

in regard to those desires. This approach assumes a standard (a norm) to be applied to those desires that might overcome the problem of deciding what action to take (the “interventional ambiguity”).

Desirability in regard to flooding, “...is always to be done in a decision-making context where a level of normative uncertainty is present” (lines 431-432). Normative uncertainty involves, “..what should be the case...” (line 435). There is going to be uncertainty, i.e., doubt or unsureness, about the desirability of flooding. This is the classical concern about making a value judgment as to what action to take. One might like a sound basis for the action that might be viewed as being without the possibility of error.

The problem of doubt (uncertainty) in regard to what action to take, i.e., what would be most desirable to do, is an age-old. If one could have an ideal sense, without error, of what would happen by taking a particular action, then that potential action could be viewed as desirable or undesirable. In the 19th century the famous English mathematician William Kingdom Clifford addressed this problem as follows (see Clifford, 1946):

Remember, then, that scientific thought is the guide of action; that the truth at which it arrives is not that which we can ideally contemplate without error, but that which we may act upon without fear.

So, how does one find, “...that which we can act upon without fear.” In other words, how can the doubt (uncertainty) be removed? Doubt is overcome by belief, which is the antonym of doubt. Belief will be the accepting of something, in this case the desirability that a particular action will be the correct one, or at least it is proving to be worthy of overcoming the doubt (uncertainty) about that action.

However, there are multiple methods for the fixing of belief (Peirce, 1877) - the most effective of these for the long term being that of science. Here “science” does not consist of a demarcation criterion for what ought to be done, but rather science is a continuing pursuit of truth by following the consequences of and learning from the actions that are taken.

Similarly the meaning of any concept, e.g., “flood,” is not something that is best set by a standard. Instead, meaning consists of the effects that conceivably follow as the practical consequences of that concept’s operation in the world (Peirce, 1878). This can lead to an axiology in regard to human aspects of flooding that derives from the philosophy of pragmatism (Baker, 1994, 1998, 2007).

Philosophical Pragmatism

The philosophical application of normativity seems directed at the goal of “flood-risk management before...problems arise...” (lines 529-531). This is an engineering approach rather than a scientific one. The relevant point here is about pragmatism as a philosophy relevant to the issue of flood risk (see Baker, 2007). The philosophical pragmatist seeks clarity in the concept “flood,” but that clarity cannot be achieved in advance of the application of the concept. This philosophical perspective has been around for at least 150 years (Peirce, 1877, 1878), and it is worthy of more consideration for the kinds of issues addressed in this manuscript.

CONCLUSION

The above comments could lead to a paper providing a completely different answer to the question that is expressed in the title of this manuscript. Rather than doing that, however, the easiest course for revising the present manuscript might be to acknowledge the various limitations and caveats on the “conceptual paradigm” that have been noted above. This revision should also emphasize the very limited applicability of that paradigm to the original question, restricting the answer to “hydrometeorological riverine floods” and their various consequences for human concerns.

This inclusion of limitations on the arguments presented in the paper might also be extended to the role of normativity, since its inclusion does not provide a complete axiology (science of values). Nevertheless, whether the authors wish to consider in regard to a more flexible axiology that would embrace a pragmatic approach (e.g., Baker, 2007) is up to them. Nearly all philosophical arguments can be rationalized, and it can always be interesting to see them presented.

References Cited in This Review

Baker, V.R., 1994, Geomorphological understanding of floods: *Geomorphology*, v. 10, p. 139-156.

Baker, V.R., 1998, Hydrological understanding and societal action: *Journal of the American Water Resources Association*, v. 34, no. 4, p. 819-825.

Baker, V.R., 2007, Flood hazard science, policy and values: A pragmatist stance: *Technology in Society*, v. 29, p. 161-168.

Baker, V.R., 2020, Global megaflood paleohydrology, in Herget, J. and Fontana, A., editors, *Palaeohydrology. Geography of the Physical Environment*: Springer, Heidelberg, p. 3-28.

Baker, V.R., and Carling, P.A., 2022, Global Late Quaternary Megafloods, in Shroder, J.J.F. (Ed.), *Treatise on Geomorphology*, 2nd Edition, vol. 6. Elsevier, Academic Press, p. 832-840.

Baker, V.R., and Milton, D.J., 1974, Erosion by Catastrophic Floods on Mars and Earth: *Icarus*, v. 23, p. 27-41.

Burr, D.M., Carling, P.A., and Baker, V.R., editors, 2009, *Megaflooding on Earth and Mars*: Cambridge University Press, Cambridge, 319 p.

Clifford W.K. The common sense of the exact sciences. New York: Alfred A. Knopf; 1946. (Reprinted from Clifford, William K. 1885. *The Common Sense of the Exact Sciences*, completed by K. Pearson. London: Kegan, Paul, Trench, and Co.)

Gautam, S., Park, S., Yu, D.J., Garcia, M., Sivapalan, M., and Shin, H.C., 2025, Homo Juridicus, Homo Heuristicus, and Homo Anticipans: A Sociohydrological Study of Operator Behavior and Flood-Drought Tradeoffs in Reservoirs: *Water Resources Research*, v. 61, issue 11, e2024WR039447 <https://doi.org/10.1029/2024WR039447>

Peirce, C. S., 1877, The fixation of belief: *Popular Science Monthly*, v. 12, p. 1-15.

Peirce, C.S., 1878, How to make our ideas clear: *Popular Science Monthly*, v. 12, p. 388-410.

Reviewer #3 (Remarks to the Author):

This is a great paper that brings much needed clarity to the concept of "flood". It is well written and well-argued, and is accessible to those who might not have technical expertise in hydrology. I also want to applaud the authors for bringing philosophical analysis to a challenge in hydrology and for presenting a strong example of how these two seemingly disparate disciplines can benefit from one another's perspectives. I recommend publication, as I think the content of the paper adds value to the community.

But, I do suggest two minor revisions, one of which is not necessary to include, but I think would articulate the additional impact/importance of the discussion.

1) (Necessary) One thing that surprised me in reading this was the lack of explicit connection between normativity and values. The values in science literature is extensive, and essentially, what this paper is getting at is that decisions made about how to define a "flood" is dependent on context, and more specifically, the values that are operating in that context. While I don't mind the use of "normativity" (this is a unique dimension this discussion adds, which I appreciate) as opposed to explicitly talking about "value-ladenness" I do think there needs to be a note early on to the effect of "By normativity, we refer to the evaluative judgments that underly determinations of what ought to be—judgments that necessarily involve values about desirable states of affairs in certain locations at certain points in time". And then include a reference early on to Elliott.

You should include this in the third introduction paragraph—somewhere in this portion "Here, we argue that utilizing normative aspects, adopting a notion of the way things ought to or should be [4], is necessary for any conceptualization of flood, however, this currently remains tacit. Therefore, any comprehensive demarcation framework for the flood phenomenon must explicitly recognize the role of this normativity." This is essentially a statement about the value-ladenness of the way different researchers/stakeholders etc. might conceptualize a "flood". The authors need to make this connection and reference some overview of this history of the discussions of how values impact scientific practice from the philosophy of science.

2. (Unnecessary) The conceptual clarity that this paper provides does some critical work for making flood research more usable/actionable: a) when researchers and stakeholders engage in the explicit identification of normative commitments (desirability, thresholds, etc.) to build the definition of a flood for a specific context, the resulting research presumably becomes more salient (relevant) to specific decision contexts, even with the potential for interventional ambiguity. Communities are better able to assess, when these components of the conceptual space are explicitly articulated, to better assess whether the implicit values in a flood model or dataset align with their own values and priorities, and it makes value deliberation and assessment of the usability of research easier. In this way it also has impacts for legitimacy of research by making value explicit and subject to democratic deliberation. Values, or normativity, as part of the conceptual process, are no longer "smuggled in" as objective technical choices, so these normative components can be objects of deliberation, making the evaluation of the legitimacy (component of usability) easier.

You could include something in the conclusion like "By explicating the normative components that fix flood concepts in specific contexts, researchers can produce work that is not only scientifically rigorous but also more transparent about its value assumptions. This transparency can serve multiple purposes: it increased the salience of researcher to particular decision contexts by making clear which values the research embeds; it enhances legitimacy by allowing stakeholders to evaluate whether those values align with community priorities and allow them to function as objects of democratic deliberation. Conceptual clarity in hydrology about the value-ladenness of how we define a flood is not just philosophically interesting, but a step towards making research more usable and actionable for responding to climate hazards". And I would cite something like "Usability of climate information: Toward a new scientific framework".

** Visit Nature Portfolio's author and referees' website at www.nature.com/authors for information about policies, services and author benefits**

Communications Earth & Environment is committed to improving transparency in authorship. As part of our efforts in this direction, we are now requesting that all authors identified as 'corresponding author' create and link their Open Researcher and Contributor Identifier (ORCID) with their account on the Manuscript Tracking System prior to acceptance. ORCID helps the scientific community achieve unambiguous attribution of all scholarly contributions. You can create and link your ORCID from the home page of the Manuscript Tracking System by clicking on 'Modify my Springer Nature account' and following the instructions in the link below. Please also inform all co-authors that they can add their ORCIDs to their accounts and that they must do so prior to acceptance.

If you experience problems in linking your ORCID, please contact the Platform Support Helpdesk.

Version 1:

Decision Letter:

Dear Dr Majszak,

Thank you for submitting your revised manuscript titled, "What is a flood?" We are delighted to say that we are happy, in principle, to publish a suitably revised version in Communications Earth & Environment.

We therefore invite you to revise your paper one last time to address remaining editorial concerns and to comply with our format requirements and to maximise the accessibility and therefore the impact of your work.

EDITORIAL REQUESTS:

Please avoid using italics for emphasis.

We recommend more declarative and informative section headers. For instance:

Intro -> Why flood definitions need clearer criteria

Physical Perspectives on Floods -> How physical factors define floods

Social and Historical Perspectives on Floods -> How human values shape the meaning of floods

*****Please take care to match our formatting and policy requirements. We will check revised manuscript and return manuscripts that do not comply. Such requests will lead to delays. *****

SUBMISSION INFORMATION:

In order to accept your paper, we require the files listed at the end of the Editorial Requests Table; the list of required files is also available at <https://www.nature.com/documents/commsj-file-checklist.pdf> .

OPEN ACCESS:

Communications Earth & Environment is a fully open access journal. Articles are made freely accessible on publication. For further information about article processing charges, open access funding, and advice and support from Nature Portfolio, please visit <https://www.nature.com/commsenv/open-access>

At acceptance, you will be provided with instructions for completing the open access licence agreement on behalf of all authors. This grants us the necessary permissions to publish your paper. Additionally, you will be asked to declare that all required third party permissions have been obtained, and to provide billing information in order to pay the article-processing

charge (APC).

Link Redacted

Best regards,

C. Kendra Gotangco Gonzales, PhD
Editorial Board Member
Communications Earth & Environment
orcid.org/0000-0002-3436-9813

Yann Benetreau, PhD
Consulting Editor, Communications Earth & Environment
Deputy Editor, Communications Sustainability
Nature Portfolio
NY office

** Visit Nature Portfolio's author and referees' website at www.nature.com/authors for information about policies, services and author benefits**

Review of “What is a flood?” Reviewed by Victor R. Baker, Dept. of Hydrology and Water Resources, The University of Arizona, Tucson, AZ. 85721 baker@arizona.edu

GENERAL COMMENTS

This is a philosophical paper in the tradition of “providing lawyer-like answers to child-like questions.” Philosophy can raise questions about fundamental concepts that may be assumed without considering the consequences of their application. The manuscript under review addresses concerns about the concept “flood” in diverse applications of that concept. It advocates a couple of demarcations, that by definition fix the boundaries or limits on what it is to be a flood.

There is an important and relevant motivation for this project, as follows, “...our proposed **demarcation** hopes to foster critical questions around the conceptual inertia in flood-risk management before...problems arise..” (lines 529-531). By developing what is proposed to be a demarcation between **physical** and **anthropocentric** floods, “...one can now ask whether what we are managing are even physical floods to begin with...” (lines 531-532). What the authors term “this broader conceptual paradigm” will avoid, “...calling all confrontations with water a de facto *flood*,” and “allow for additional approaches, such as **anthropocentric** approaches...to be undertaken for flood risk management...” (lines 535-539).

The problem to be solved by this demarcation project is termed “interventional ambiguity,” which is, “...the lack of consensus in communities...on whether to seek engineered solutions to specific flood-related problems...” (lines 27-29) The manuscript does not claim to resolve this issue. Instead, it hopes that the perspective that it provides, “...fosters dialogue within the **hydroclimatology** community on how best to conceptualize this phenomenon, which could lead to further refinement of this proposed demarcation framework” (lines 559-562).

The proposed “conceptual paradigm” to deal with the problem of “interventional ambiguity” is the recognition of a need for “...conceptual clarity...[in the] use of the term and the impacts of the events in question...” (lines 52-54). The problem of the term “flood” referring to different “conceptions of the phenomenon” is to be resolved by incorporating “normativity” (*what ought to be*) in regard to both the physical and the anthropocentric answers to question about the meaning of word “flood.” **Physical floods** are defined as, “...pools and fluxes beyond a **normatively-specified** rarity threshold – the exceedence probability” (lines 22-23). “**Anthropocentric floods** are then physical floods deemed desirable or undesirable” (Line 25).

Normativity determines *what ought to be* in regard to a **norm**, that is in regard to some standard. For the physical floods this standard is determined by “a mathematical definition,” the exceedence probability, which given by an equation. While this is claimed to be a physics-based standard, it seems to be more to be an engineering criterion used for evaluate risk, which would be the multiplication of this probability standard times a consequence.

One problem here is that many physical floods do not have probabilities because they are singular events. Constructed dam failures occur because of design defects, or even human negligence. Extreme high-energy floods may occur because of unique circumstances that have nothing to do with a time sequence of river discharges, storm events, or other phenomena subject to estimates of their likelihoods. The proposed norm does seem applicable to the hydroclimatological riverine flooding phenomena (a subset of “what is a flood?”) that is the apparent focus of this manuscript.

For the antropocentric floods the normative standard is to be a value judgment as to whether a flood is desirable or undesirable. This achieves a demarcation, which is a fixing of the boundaries or limits on something, in this case, the concept of a “flood.” As noted above, this value judgment is directed making management decisions “before...problems arise.” This notion of making a prior judgment is also an engineering-oriented criterion that needs to be distinguished from a pragmatic/scientific approach that will be noted below.

DETAILED COMMENTS

The Very Limited Conception of “Flood”

The manuscript should not imply that it is answering a very general question concerning the concept “flood.” It is actually dealing with very limited aspects of that phenomenon. It does not deal at all with the reality that ancient massive floods have characterized both Earth and Mars (Baker and Milton, 1974; Burr et al., 2009; Baker, 2020).

It also does not deal with man-made (i.e., “human” floods that would involve a different wrinkle on what is subject to the demarcation: “anthropocentric floods.” Famous examples are the many dam failure floods that involve human negligence. There is a rather large literature on these (multiple books) with detailed case studies, There are also examples of massive flooding being used as a weapon of war, as occurred in northern China during the 1930 (leading to massive losses of life and the displacement of millions of people). Flooding caused by malicious human intent includes past Mississippi River levee ruptures in Black-populated parrishes upstream of New Orelans that were purposely perpetrated to reduce river flood stages that might otherwise damage white-owned business interests in that city.

The “physical floods” demarcation seems to ignore seasonal inundations, which occur every year. The concept of exceedance probability is irrelevant to such “floods.”

There clearly are important human aspects of the risks posed by tsunami floods, but these are not susceptible to being considered “desirable” relative to human interests. There is no ambiguity whether such floods are desirable or undesirable; they are always undesirable in regard to humans that are placed at risk.

Indeed the emphasis on exceedance probabilities and other comments in the text indicate that the physical flooding focus of this manuscript is on engineering issues related to hydroclimatic riverine flooding, rather than on scientific concerns that might othwise be suggested by the title of the manuscript. More will be said on this below.

Geological Perspective – lines 112-135

This section is limited in its demarcation of what it means to be “geological.” As a geologist, I find the whole section to be so misleading that it might be best to just eliminate all of it, including Figure 1.

This section of the text really does not add to the main objective of the paper, which, as noted above, concerns hydroclimatic riverine floods, not floods in general. Eliminating this section would shorten the manuscript, and it would also focus it more on the what seems to be the main objective.

For some a more general view on how geological thinking can relate to flooding, including its “anthropocentric” aspects, one can see Baker (1994).

Though eliminating the section is the easiest path to revision, acknowledging the limited aspect of its whole approach – that it considers concept “flood” largely from an engineering viewpoint – would be helpful to the reader.

Figure 1 - Distribution of archaic humans, prehistoric sites, and ancient civilizations on active global floodplains and flood-related landforms.

The base map for this figure (Panel A) is inappropriate. It emphasizes global relief, much of which is submarine - having nothing to do with the “flooding” that is of concern in the manuscript. Most of the elevation scale scale is devoted to submarine relief (0 to -11 km depth).

The legend refers of “prehistoric outburst floods” and supposedly shows locations with blue dots. However, there is only one blue dot that I can see on the map, and it is on the Arabian Peninsula where I know of no outburst flood evidence. Actually there are scores of ancient outburst floods that have been documented and mapped on multiple land areas, mainly near the margins of the large Pleistocene ice sheets that formerly covered parts of Eurasia, North and South America (e.g., Baker, 2020; Baker and Carling, 2022).

The locations of “Holocene Human Floods” are to be indicated as large purple dots (from the caption), but they seem to be shown as small reddish dots on the map. Also, the concept of “Holocene Human Floods” is not defined in the text, which focuses on “anthropocentric floods” defined in terms of their disirability or lack there of. During the Holocene there were humans all over the land areas of the map (except Antarctica).

The distribution of “Archaic humans” is to be designated by small brown dots according to the caption, but the assumed distribution is shown by large brown dots on the map. The legend for “human presence” defines “Archaic” as between 2 Ma and 30 ka. However, humans are now known to have arrived in Australia by 40-50 ka. There is also mounting evidence suggesting humans in the Americas prior to 30 ka, and humans have long been in southern Africa.

Finally the distribution of “global floodplains” is certainly not shown globally. There are

many floodplains in northern North America and northern Eurasia. The distribution seems to be arbitrarily cut off at a latitude of 60 degrees north. Many of what are mapped as “floodplains” are actually be seasonal wetlands. Moreover, the high density of “Holocene floods” obscures the the pattern of floodplains in many of the areas where “anthropocentric floods” would be of most interest, as in western Europe, eastern China, and the eastern U.S.

There are so many problems with this map that it would be simplest to just eliminate it along with the relevant text.

Physical floods – defined as “...pools and fluxes beyond a normatively-specified rarity threshold – the exceedance probability”

“Pools and fluxes” characterizes physical phenomena that are not floods. Oceans are pools of water that have huge fluxes (ocean currents). These pools and fluxes are physical, and they have nothing to do with exceedance probabilities.

There are also floods, known as megafloods, with fluxes that are measured in the same units (sverdrups – millions of cubic meters of water per second) as ocean currents (e.g., Baker and Carling, 2022). There are also cases when oceans (pools) can generate fluxes that produce megafloods over the land. This occurred 5.33 million years ago when the Atlantic Ocean spilled over the divide that previously separated what is now Spain and Morocco. The resulting “Zanclean Flood” created what is now the Strait of Gibraltar as it spilled into the then-dry Mediterranean Basin. What is now the English Channel was created by mega-flooding. This also occurred with the straits connecting what is now the Black Sea to the Mediterraneanian.

These various megafloods are definitely physical, but the concept of exceedance probability has nothing to do with them. Exceedance probabilities are used in engineering evaluations of risk (consequence time probability) for purposes of design, mitigation works, etc. Extremely large physical floods are essentially unique events for which the concept of probability is irrelevant. There is no quantitative measure of expectation for the likelihood that such a future event will occur because the circumstances that produce these events are unique to the special circumstances of the events themselves. The problem is not analogous to the counting of colored beans in a jar. There is only the certainty that what has happened can indeed happen.

I could go on at great length about ancient floods on the planet Mars and much else, but the main conclusion here is that the whole argument of this manuscript is best limited to hydroclimatic riverine floods.

Anthropocentric floods – defined as “physical floods deemed desirable or undesirable”

The complexity of human decisions that have to be made in regard to flooding are well illustrated by the very recent paper on where formal rule curves intersect with human discretionary judgment in regard to reservoir operations relative to flood risk (Gautam et

al., 2025). It would be well to check how the demarcation of “anthropocentric floods” fits with this aspect of flood hydrology.

Normativity

Normativity is only one approach to the general issue of values. An alternative approach would be to consider the complete axiology of the situation (e.g., Baker, 1998).

As noted in the discussion of “interventional ambiguity” the focus of this manuscript is what is desirable or undesirable in regard to the physical phenomenon of (hydroclimatic riverine) flooding. It is claimed (lines 363-364.) that, “...normativity is specifically regarding adopting a notion of desirability for the presence of water at the identified time and location.” In other words, what an individual wants or wishes for (desires) as a course action is to be specified by *what ought to be* (normativity) in regard to those desires. This approach assumes a standard (a norm) to be applied to those desires that might overcome the problem of deciding what action to take (the “interventional ambiguity”).

Desirability in regard to flooding, “...is always to be done in a decision-making context where a level of normative uncertainty is present” (lines 431-432). Normative uncertainty involves, “..what should be the case...” (line 435). There is going to be uncertainty, i.e., **doubt** or unsureness, about the desirability of flooding. This is the classical concern about making a value judgment as to what action to take. One might like a sound basis for the action that might be viewed as being without the possibility of error

The problem of doubt (uncertainty) in regard to what action to take, i.e., what would be most desirable to do, is an age-old. If one could have an ideal sense, without error, of what would happen by taking a particular action, then that potential action could be viewed as desirable or undesirable. The famous English mathematician William Kingdom Clifford addressed this problem as follows:

Remember, then, that scientific thought is the guide of action; that the truth at which it arrives is not that which we can ideally contemplate without error, but that which we may act upon without fear.

So, how does one find, “...that which we can act upon without fear.” In other words, how can the doubt (uncertainty) be removed? Doubt is overcome by belief, which is the antonym of doubt. Belief will be the accepting of something, in this case the desirability that a particular action will be the correct one, or at least it is proving to be worthy of overcoming the doubt (uncertainty) about that action.

However, there are multiple methods for the fixing of belief (Peirce, 1877) - the most effective of these for the long term being that of science. Here “science” does not consist of a demarcation criterion for what ought to be done, but rather science is a continuing pursuit of truth by following the consequences of and learning from the actions that are taken.

Similarly the meaning of any concept, e.g., “flood,” is not something that is best set by a standard. Instead, meaning consists of the effects that conceivably follow as the practical

consequences of that concept's operation in the world (Peirce, 1878). This can lead to an axiology in regard to human aspects of flooding that derives from the philosophy of pragmatism (Baker, 1994, 1998, 2007).

Philosophical Pragmatism

The philosophical application of normativity seems directed at the goal of “flood-risk management before...problems arise...” (lines 529-531). This is an engineering approach rather than a scientific one. The relevant point here is about pragmatism as a philosophy relevant to the issue of flood risk (see Baker, 2007). The philosophical pragmatist seeks clarity in the concept “flood,” but that clarity cannot be achieved in advance of the application of the concept. This philosophical perspective has been around for at least 150 years (Peirce, 1877, 1878), and it is worthy of more consideration for the kinds of issues addressed in this manuscript.

CONCLUSION

The above comments could lead to a paper providing a completely different answer to the question that is expressed in the title of this manuscript. Rather than doing that, however, the easiest course for revising the present manuscript might be to acknowledge the various limitations and caveats on the “conceptual paradigm” that have been noted above. This revision should also emphasize the very limited applicability of that paradigm to the original question, restricting the answer to “hydrometeorological riverine floods” and their various consequences for human concerns.

This inclusion of limitations on the arguments presented in the paper might also be extended to the role of normativity, since its inclusion does not provide a complete axiology (science of values). Nevertheless, whether the authors wish to consider in regard to a more flexible axiology that would embrace a pragmatic approach (e.g., Baker, 2007) is up to them. Nearly all philosophical arguments can be rationalized, and it can always be interesting to see them presented.

References Cited in This Review

Baker, V.R., 1994, Geomorphological understanding of floods: *Geomorphology*, v. 10, p. 139-156.

Baker, V.R., 1998, Hydrological understanding and societal action: *Journal of the American Water Resources Association*, v. 34, no. 4, p. 819-825.

Baker, V.R., 2007, Flood hazard science, policy and values: A pragmatist stance: *Technology in Society*, v. 29, p. 161-168.

Baker, V.R., 2020, Global megaflood paleohydrology, in Herget, J. and Fontana, A., editors, *Palaeohydrology. Geography of the Physical Environment*: Springer, Heidelberg, p. 3-28.

Baker, V.R., and Carling, P.A., 2022, Global Late Quaternary Megafloods, in Shroder, J.J.F. (Ed.), *Treatise on Geomorphology*, 2nd Edition, vol. 6. Elsevier, Academic Press, p. 832-840.

Baker, V.R., and Milton, D.J., 1974, Erosion by Catastrophic Floods on Mars and Earth: *Icarus*, v. 23, p. 27-41.

Burr, D.M., Carling, P.A., and Baker, V.R., editors, 2009, Megaflooding on Earth and Mars: Cambridge University Press, Cambridge, 319 p.

Clifford WK. The common sense of the exact sciences. New York: Alfred A. Knopf; 1946. (Reprinted from Clifford, William K. 1885. *The Common Sense of the Exact Sciences*, completed by K. Pearson. London: Kegan, Paul, Trench, and Co.)

Gautam, S., Park, S., Yu, D.J., Garcia, M., Sivapalan, M., and Shin, H.C., 2025, *Homo Juridicus, Homo Heuristicus, and Homo Anticipans: A Sociohydrological Study of Operator Behavior and Flood-Drought Tradeoffs in Reservoirs*: *Water Resources Research*, v. 61, issue 11, e2024WR039447 <https://doi.org/10.1029/2024WR039447>

Peirce, C. S., 1877, The fixation of belief: *Popular Science Monthly*, v. 12, p. 1-15.

Peirce, C.S., 1878, How to make our ideas clear: *Popular Science Monthly*, v. 12, p. 388-410.

Dear Editor and Reviewers,

We would like to thank you all for these helpful and extensive reviews. We are grateful for the time and effort you have given us in reviewing our manuscript, and we have significantly revised the manuscript following the remarks of all three reviewers. All changes in the document, as well as our responses to the reviewer's comments below, are marked in **BLUE**. We look forward to your subsequent reviews of this new version of the manuscript and any comments you can provide to further improve the manuscript.

Replies to the referees,

Reviewer #1 (Remarks to the Author):

The subject of the Perspective by Majszak et al. and the concepts discussed are important and timely. The question, “What is a flood”, is thought-provoking, because I think that many of us, as the authors suggest, just take a flood to be an objectively determined physical event. Their framing of floods as having a normative component relating to what individuals subjectively think about their situation (e.g., location and timeframe) seems sound to me based on my work. The authors might find the following to be relevant to their research, especially regarding timeframes and risk tolerance/aversion:

- Lutz, T., 2024, A Review of the Philosophy of Flood Risk Communication and Education: New Insights Informed by Critical Complexity: *Water* 16 (23) 3459, <https://doi.org/10.3390/w16233459>

Thank you for recommending this paper, we agree that the points discussed in this review are important and relevant to the discussion we are having in this perspective. We have included this work within the manuscript. A reference to this work can be found at the end of Section 4 when we discuss interventional ambiguity in greater detail and other aspects of management related to anthropocentric floods.

The Perspective is organized straightforwardly to bring out the idea that a sound conception of floods requires a normative component: Section 2 (Perspective on Floods), Section 3 (... Socio-physical conceptualization...), and Section 4 (Interventional Ambiguity) form the heart of the paper. “Ambiguity” is key because failures in flood management are often seen as technical problems, to be addressed by better engineering or better communication of status quo ideas (e.g., Q100). The authors recognize that in complex systems (e.g., societies) there is always an incompleteness or ambiguity that needs to be considered.

Despite all the good things to be said about the paper, I think substantial rewriting should be required before publication occurs. I was driven to distraction by several things.

- Inefficient writing slows down and burdens the reader. More than 1 of every 7 lines contains a sentence beginning with an unnecessary or vague word or phrase such as “This” (26x), “Here” (17x), “Thus” (7x), “In turn” (5x), “In this way” (29x). Usually such wasted words can be eliminated by simply writing clearly in the first place so that it doesn’t become the reader’s job to figure out what the author is trying to say.
- The paper’s subject requires the words “normative” and “should” to be used a bit. But the fact that they appear 105 times in the text (once in every 5 lines, on average) is mind-numbing for the reader. It makes one wonder what’s really new and what’s just repetitive. I think the authors may be trying to make sure the reader is catching on to the novelty of their ideas but they are overdoing it! The paper can be written less repetitively.

We would like to thank the reviewer for this comment, we agree that the language we have used in the previous draft may be burdensome to the reader. We have completed a substantial rewriting of the document in line with the suggestions from this reviewer. This has resulted in removing the vast majority of instances where the troublesome terms/phrases highlighted by the reviewer (i.e. “This”, “Here”, “Thus”, “In turn”, “In this way”) were used to start a sentence. The counts for these words to start sentences are as follows: “This” 15x, “Here” 2x, “Thus” 5x, “In turn” 1x, “In this way” 1x. Additionally, great care was taken to limit the repetitiveness of other key words, such as “normative”/“normatively” and “should”, which were overused in the initial version of this perspective paper; they now appear only 25x and 30x, respectively (or once every in every 10 lines, on average). By performing these alterations, we also significantly reduced repetitive sentences and ideas throughout the paper.

The figures have their own issues.

- Captions for Figures 2 & 3 go beyond describing the figure to include explanation that should be in the text.

We agree with this comment. With this in mind, we have reduced the length of Figure 2, and included the description of Figure 3 into the main text (as Figure 3 has been removed).

- I question whether Figure 1, which is complicated, contributes essential information. I think the importance of timeframes and locations is adequately communicated in the text; this figure could be eliminated without affecting the impact of the paper.

We agree with the reviewer, and have removed Figure 1 entirely from this manuscript.

- Figure 2 is inconsistent and confusing. If normative influence increases upward (arrow on left) why do the ‘Physical’ and ‘Pools and Fluxes’ boxes extend above the Anthropocentric box? Why is there a gradational Clear-Fuzzy bar on the right when there is essentially no gradation shown in the figure? And, why is this concept illustrated as 2-D? The horizontal direction has no axis and seems to be there only to turn a 1-D concept into 2-D diagram.

We want to first thank the reviewer for this helpful comment. We have redesigned Figure 2 (now labeled Figure 1) with these ideas in mind. In short, we have adjusted the figure to remove the space between the three sides of the boxes, where this was only meant to highlight that these concepts are nested, not that the normative influence extends into “Pools and Fluxes”. We have also shortened the arrow of “normative influence” to end at the top of the “Anthropocentric Floods” box to reflect this point as well. We have also increased the contrast within the gradation, to improve the readability of the gradation across the “Anthropocentric Floods” box.

Regarding the dimensionality of the figure, while there is only one dimension (the vertical impact of “normative influence”) which impacts the demarcation of the flood concept, the figure is shown as a series of boxes to allow for the representation of two aspects. (1) the nested aspect of the flood concept, where all “Anthropocentric Floods” are examples of “Physical Floods” and all “Physical Floods” are examples of “Pools and Fluxes”, and (2) that within this conceptual space there is interventional ambiguity, where there is a range of this ambiguity. To show the range of ambiguity a gradient pattern must be used as this is not a binary concept. The horizontal display of this gradient pattern was used as this variability is most clearly visually represented across a box so as not to confuse the range of interventional ambiguity with the vertical change in the influence of normativity. Finally, we also want to be clear and state that this figure is meant only as a schematic representation of the conceptual space, not as any formal set-theoretic space across an x and y-axis.

- Figure 3 is inconsistent and confusing, too. Shouldn’t each of the three sub-diagrams have their own set of axes? Or, why do the A.F. boxes inside have different sizes and locations within each physical flood box? More substantially, to clarify the text for the reader, I think that the hypothetical values should be replaced by real-world examples. Otherwise, I don’t think it’s clear what the “values” might be. Since the attributes of the three individuals are given in the text (lines 390-396), why not make the diagram more explicit?

We completely agree with the reviewer and thank them for bringing this to our attention. This figure was attempting to show the impact of the adoption (or lack of adoption) of values on three individuals. However, the conceptual space we constructed in Figure 2 is not conducive, for many of the reasons highlighted by this reviewer, to represent these changes on a two axis

mapping. We have omitted this figure as a result and have elected to leave this discussion to occur only within the manuscript itself.

To recap: This paper brings forth interesting and important insights into a “simple” question: What is a flood? Many in the readership may be led to expand their thinking about flood risk and flood communication, especially with regard to interventional ambiguity. However, whether readers are able to get the intended ideas depends on improving the writing and figures so that the message gets across.

Once again, we would like to thank the reviewer for this review. We feel that the manuscript has improved in both readability and ease of understanding due to the comments highlighted and outlined by this review.

Reviewer #2 (Remarks to the Author):

Review of “What is a flood? Reviewed by Victor R. Baker, Dept. of Hydrology and Water Resources, The University of Arizona, Tucson, AZ. 85721 baker@arizona.edu

GENERAL COMMENTS

This is a philosophical paper in the tradition of “providing lawyer-like answers to child-like questions.” Philosophy can raise questions about fundamental concepts that may be assumed without considering the consequences of their application. The manuscript under review addresses concerns about the concept “flood” in diverse applications of that concept. It advocates a couple of demarcations, that by definition fix the boundaries or limits on what it is to be a flood.

There is an important and relevant motivation for this project, as follows, “...our proposed demarcation hopes to foster critical questions around the conceptual inertia in flood-risk management before...problems arise..” (lines 529-531). By developing what is proposed to be a demarcation between physical and anthropocentric floods, “...one can now ask whether what we are managing are even physical floods to begin with...” (lines 531-532). What the authors term “this broader conceptual paradigm” will avoid, “...calling all confrontations with water a de facto flood,” and “allow for additional approaches, such as anthropocentric approaches...to be undertaken for flood risk management...” (lines 535-539).

We thank Prof. Baker for the careful reading and constructive comments, which helped strengthen the revised manuscript. In response, we have made substantial changes to the manuscript. We clarified the framing and scope of the paper, explicitly acknowledged key limitations of the proposed demarcation framework, and removed under-developed sections and figures that were not essential to the main argument. We also added a dedicated “challenges and outlook” section to clarify applicability and to outline productive directions for future work. Please find below our detailed response to each of your comments.

The problem to be solved by this demarcation project is termed “interventional ambiguity,” which is, “...the lack of consensus in communities...on whether to seek engineered solutions to specific flood-related problems...” (lines 27-29) The manuscript does not claim to resolve this issue. Instead, it hopes that the perspective that it provides, “...fosters dialogue within the hydroclimatology community on how best to conceptualize this phenomenon, which could lead to further refinement of this proposed demarcation framework” (lines 559-562).

We thank you for pointing this out and agree with what has been written above. The manuscript does not claim to resolve interventional ambiguity or to prescribe engineered solutions. Our aim is to clarify why interventional ambiguity arises (and why it can persist even when the physical event is agreed upon) and to offer a common conceptual vocabulary that can foster dialogue and motivate future refinement of the framework. To avoid over-claiming, we made this scope explicit in several places in Section 4.

The proposed “conceptual paradigm” to deal with the problem of “interventional ambiguity” is the recognition of a need for “...conceptual clarity...[in the] use of the term and the impacts of the events in question...” (lines 52-54). The problem of the term “flood” referring to different “conceptions of the phenomenon” is to be resolved by incorporating “normativity” (what ought to be) in regard to both the physical and the anthropocentric answers to question about the meaning of word “flood.” Physical floods are defined as, “...pools and fluxes beyond a normatively-specified rarity threshold – the exceedance probability” (lines 22-23). “Anthropocentric floods are then physical floods deemed desirable or undesirable” (Line 25).

We agree with Prof. Baker’s reading of these passages. Specifically, that the term “flood” is currently being used for different conceptions of the phenomenon, and that our framework is meant to make those differences explicit by showing where normativity enters: first in specifying relevance parameters and a rarity threshold for physical floods, and then in introducing desirability to distinguish anthropocentric floods. To reduce ambiguity, we clarified these definitions and the role of normativity in several places throughout the manuscript (including the Abstract and Section 3).

Normativity determines what ought to be in regard to a norm, that is in regard to some standard. For the physical floods this standard is determined by “a mathematical definition,” the exceedance probability, which given by an equation. While this is claimed to be a physics-based standard, it seems to be more to be an engineering criterion used for evaluate risk, which would be the multiplication of this probability standard times a consequence.

One problem here is that many physical floods do not have probabilities because they are singular events. Constructed dam failures occur because of design defects, or even human negligence. Extreme high-energy floods may occur because of unique circumstances that have nothing to do with a time sequence of river discharges, storm events, or other phenomena subject to estimates of their likelihoods. The proposed norm does seem applicable to the hydroclimatological riverine flooding phenomena (a subset of “what is a flood?”) that is the apparent focus of this manuscript.

Thank you very much for this great comment. We agree with this concern. An exceedance-probability formulation is most naturally suited to hydroclimatological riverine

flooding, where event magnitudes can be related to a stochastic process and adopting a return period for the rarity threshold is meaningful. We now acknowledge explicitly within the manuscript that some floods—such as dam failures driven by design defects/negligence, or highly contingent “one-off” high-energy events—are not well represented by a stationary frequency interpretation. We discuss these cases within the new Sec. 5 on potential challenges and further outlook. This has allowed us to emphasize that the rarity of these high-energetic floods can be treated as a broader end-member concept or within a Bayesian conception of rarity, rather than only a frequentist probability from a long time series. In this section we now clarify that the normatively defined rarity threshold necessary for the identification of a physical flood can be done in ways other than the adoption of a return period. This has greatly improved the ability of our framework to capture these unique occurrences or “singular events” and we thank Prof. Baker for bringing this to our attention.

For the anthropocentric floods the normative standard is to be a value judgment as to whether a flood is desirable or undesirable. This achieves a demarcation, which is a fixing of the boundaries or limits on something, in this case, the concept of a “flood.” As noted above, this value judgment is directed making management decisions “before...problems arise.” This notion of making a prior judgment is also an engineering-oriented criterion that needs to be distinguished from a pragmatic/scientific approach that will be noted below.

DETAILED COMMENTS

The Very Limited Conception of “Flood”

The manuscript should not imply that it is answering a very general question concerning the concept “flood.” It is actually dealing with very limited aspects of that phenomenon. It does not deal at all with the reality that ancient massive floods have characterized both Earth and Mars (Baker and Milton, 1974; Burr et al., 2009; Baker, 2020).

We thank Prof. Baker for this comment. We agree that the earlier version of this manuscript does not do a sufficient job of discussing these unique/catastrophic aspects of the flood phenomenon. We also agree that a complete conceptual picture should acknowledge that rare, high-magnitude floods recorded in geological archives on Earth and in planetary geomorphology on Mars are central to how “flood” is used in Earth and planetary sciences. To address this, we refined a few paragraphs about the terrestrial and Martian megafloods within the manuscript as end-member physical floods characterized by extreme magnitude and rarity and by relevance parameters (timescales and spatial scales) that differ from modern hydroclimatological usage. These end-members help motivate why our framework foregrounds the choice of relevance parameters and thresholds, while also underscoring that probabilistic/return-period language may be less

applicable for singular or poorly sampled events. With these changes we contend that our proposed framework can capture these events as being instances of physical floods.

It also does not deal with man-made (i.e., “human” floods that would involve a different wrinkle on what is subject to the demarcation: “anthropocentric floods.” Famous examples are the many dam failure floods that involve human negligence. There is a rather large literature on these (multiple books) with detailed case studies, There are also examples of massive flooding being used as a weapon of war, as occurred in northern China during the 1930 (leading to massive losses of life and the displacement of millions of people). Flooding caused by malicious human intent includes past Mississippi River levee ruptures in Black-populated parishes upstream of New Orleans that were purposely perpetrated to reduce river flood stages that might otherwise damage white-owned business interests in that city.

Thank you for this valuable comment. We agree that man-made floods are not discussed within the initial version of this manuscript. However, we see that our conception of “flood” is a first order definition and would be able to handle these events as being instances of physical floods (and anthropocentric floods). Our conception of physical flood makes no reference to why or how the amount of water was present in a location at a given moment in time. One can adopt relevance parameters (the location and timescale of interest) as well as set a rarity threshold where these events highlighted above can be identified as physical floods. If one would like to then discuss the reason behind the physical flood they would need to adopt a second order concept and say these are *human induced physical floods* or, more precisely, these are *human induced anthropocentric floods*. We discuss, briefly, the impact of our framework on identifying second order floods in Section 3.1, in terms of coastal physical flood events.

The “physical floods” demarcation seems to ignore seasonal inundations, which occur every year. The concept of exceedance probability is irrelevant to such “floods.”

We thank Prof. Baker for this comment and agree that we did not explicitly discuss how seasonal inundation could fit into our proposed demarcation framework. We have added a new paragraph discussing this potential issue in the new section “Potential challenges and outlook”. In this new paragraph we argue that our proposed framework can address seasonal inundation, where the normatively defined relevance parameters and rarity threshold can classify this inundation as a physical flooding (or not) depending on how these parameters and the threshold are set.

There clearly are important human aspects of the risks posed by tsunami floods, but these are not susceptible to being considered “desirable” relative to human interests. There is no ambiguity whether such floods are desirable or undesirable; they are always undesirable in regard to humans that are placed at risk.

We thank Prof. Baker for this comment and bringing this tsunami example to our attention. To be clear, we don't necessarily see this example as being problematic or against our demarcation framework. Our account does allow for some anthropocentric floods to be interventionally clear, where some tsunamis may fall into this category if they are, as Prof. Baker has suggested, "always undesirable in regard to humans that are placed at risk". However, we have used this example to highlight a unique aspect of the expression of values in decision-making in these moments of potential catastrophe. We have added this example to our discussion of interventional ambiguity in Section 4. We state:

"There are some examples, such as protection of human life, which are widespread across societies and can, in some sense, be seen as widely agreed upon. Instances of tsunami inundation may be argued as such an example of an anthropocentric flood which is universally undesirable. While the widespread agreement on the undesirability of tsunami inundation may be true, relative to those individuals experiencing the inundation caused by the tsunami, such agreement on the adoption and utilization of the value of human life cannot be universally guaranteed. Take for example the case of COVID19 where other values, such as monetary loss, conflicted with the values for the protection of life and public health (Bognar 2024). Thus, even for a well-established value, which may on the surface appear to be universal, the utilization of the value in decision-making is not guaranteed."

In this new section we outline how tsunamis may be viewed as undesirable for all those individuals experiencing the inundation caused by the tsunami. However, this view of the tsunamis being universally undesirable may not be true for any person who examines the situation. In this discussion we highlight that it is possible for there to be certain people who adopt other values in how they view the event. While we don't provide this example directly in the section, there could also be individuals who are not directly impacted by the event in question and may, as a result, be agnostic to the desirability of this flood. We see the discussion of desirability and interventional ambiguity to then still be relevant in these contexts.

Indeed the emphasis on exceedance probabilities and other comments in the text indicate that the physical flooding focus of this manuscript is on engineering issues related to hydroclimatic riverine flooding, rather than on scientific concerns that might otherwise be suggested by the title of the manuscript. More will be said on this below.

We would like to again thank Prof. Baker for these comments regarding the perceived limitations of our framework as they have provided us the opportunity to highlight underdiscussed aspects of our manuscript. However, given our responses to the concerns, we see our frameworks as being applicable beyond hydroclimatic riverine flooding, as it is able to capture flooding of a larger variety of second-order physical floods, many of which Prof. Baker has identified, highlighting its generalizability.

Geological Perspective – lines 112-135

This section is limited in its demarcation of what it means to be “geological.” As a geologist, I find the whole section to be so misleading that it might be best to just eliminate all of it, including Figure 1.

This section of the text really does not add to the main objective of the paper, which, as noted above, concerns hydroclimatic riverine floods, not floods in general. Eliminating this section would shorten the manuscript, and it would also focus it more on the what seems to be the main objective.

Thank you for this great comment. We deleted the figure, but strengthened the geological context. As you suggested, we now treat ancient floods that shaped Earth and Mars as useful end-members relative to more frequently occurring floods. These cases provide an additional angle for discussing how flood frequency, magnitude, and duration can vary across timescales and may change through time. Please see the revised paragraphs in the manuscript.

For some a more general view on how geological thinking can relate to flooding, including its “anthropocentric” aspects, one can see Baker (1994).

Thank you for sharing this helpful reference. We have now cited Baker (1994) and incorporated its perspective where relevant.

Though eliminating the section is the easiest path to revision, acknowledging the limited aspect of its whole approach – that it considers concept “flood” largely from an engineering viewpoint – would be helpful to the reader.

Thank you for this valuable comment, which helped us sharpen the paper. We now use the geologic discussion as well as other examples to highlight the generality of our proposed framework beyond an exclusively engineering view of flooding.

Figure 1 - Distribution of archaic humans, prehistoric sites, and ancient civilizations on active global floodplains and flood-related landforms.

The base map for this figure (Panel A) is inappropriate. It emphasizes global relief, much of which is submarine - having nothing to do with the “flooding” that is of concern in the manuscript. Most of the elevation scale scale is devoted to submarine relief (0 to -11 km depth).

Thanks for this comment. We agree, and have therefore deleted the figure.

The legend refers of “prehistoric outburst floods” and supposedly shows locations with blue dots. However, there is only one blue dot that I can see on the map, and it is on the Arabian Peninsula where I know of no outburst flood evidence. Actually there are scores of ancient outburst floods that have been documented and mapped on multiple land areas, mainly near the margins of the large Pleistocene ice sheets that formerly covered parts of Eurasia, North and South America (e.g., Baker, 2020; Baker and Carling, 2022).

The locations of “Holocene Human Floods” are to be indicated as large purple dots (from the caption), but they seem to be shown as small reddish dots on the map. Also, the concept of “Holocene Human Floods” is not defined in the text, which focuses on “anthropocentric floods” defined in terms of their desirability or lack thereof. During the Holocene there were humans all over the land areas of the map (except Antarctica).

The distribution of “Archaic humans” is to be designated by small brown dots according to the caption, but the assumed distribution is shown by large brown dots on the map. The legend for “human presence” defines “Archaic” as between 2 Ma and 30 ka. However, humans are now known to have arrived in Australia by 40-50 ka. There is also mounting evidence suggesting humans in the Americas prior to 30 ka, and humans have long been in southern Africa.

Finally the distribution of “global floodplains” is certainly not shown globally. There are many floodplains in northern North America and northern Eurasia. The distribution seems to be arbitrarily cut off at a latitude of 60 degrees north. Many of what are mapped as “floodplains” are actually be seasonal wetlands. Moreover, the high density of “Holocene floods” obscures the the pattern of floodplains in many of the areas where “anthropocentric floods” would be of most interest, as in western Europe, eastern China, and the eastern U.S.

There are so many problems with this map that it would be simplest to just eliminate it along with the relevant text.

We completely agree with Prof. Baker on these issues and we thank them for bringing these problems to our attention. As a result we have removed Figure 1 entirely from this manuscript.

Physical floods – defined as “...pools and fluxes beyond a normatively-specified rarity threshold – the exceedance probability”

“Pools and fluxes” characterizes physical phenomena that are not floods. Oceans are pools of water that have huge fluxes (ocean currents). These pools and fluxes are physical, and they have nothing to do with exceedance probabilities.

There are also floods, known as megafloods, with fluxes that are measured in the same units (sverdrups – millions of cubic meters of water per second) as ocean currents (e.g., Baker and Carling, 2022). There are also cases when oceans (pools) can generate fluxes that produce megafloods over the land. This occurred 5.33 million years ago when the Atlantic Ocean spilled over the divide that previously separated what is now Spain and Morocco. The resulting “Zanclean Flood” created what is now the Strait of Gibraltar as it spilled into the then-dry Mediterranean Basin. What is now the English Channel was created by mega-flooding. This also occurred with the straits connecting what is now the Black Sea to the Mediterranean.

These various megafloods are definitely physical, but the concept of exceedance probability has nothing to do with them. Exceedance probabilities are used in engineering evaluations of risk (consequence time probability) for purposes of design, mitigation works, etc. Extremely large physical floods are essentially unique events for which the concept of probability is irrelevant. There is no quantitative measure of expectation for the likelihood that such a future event will occur because the circumstances that produce these events are unique to the special circumstances of the events themselves. The problem is not analogous to the counting of colored beans in a jar. There is only the certainty that what has happened can indeed happen.

I could go on at great length about ancient floods on the planet Mars and much else, but the main conclusion here is that the whole argument of this manuscript is best limited to hydroclimatic riverine floods.

We thank Prof. Baker for these helpful comments and examples, which show how rare, high-magnitude events can reset geomorphic boundary conditions regionally and at a planet-scale. We agree with the critique. Megafloods and other highly contingent events are clearly physical floods, but they are not naturally captured by a stationary, frequency-based exceedance-probability framing. We therefore revised the manuscript to make two points explicit.

First, exceedance probability is not a requirement for physical floods: it is an operational, hydroclimatology-oriented way to express a rarity threshold for the riverine hydroclimatic end-member, where probabilistic descriptions from time series are meaningful and commonly used. However, we now discuss that the rarity threshold can be set with a different conception, beyond a return period, specifically with a Bayesian conception of rarity. This is done in the new Section. 5, when discussing some potential challenges to our framework.

Second, megafloods and ocean-spillover floods represent a different end-member: essentially singular/contingent events for which return periods are not well defined. For these cases, we do not argue that a quantitative exceedance probability exists; instead, the relevant notion of “rarity”

is qualitative/epistemic—i.e., exceptionalness relative to an explicit baseline or knowledge state—while acknowledging that precise likelihoods are often poorly constrained. Finally, we agree that the manuscript reads most cleanly when framed around hydroclimatic riverine floods, and we revised the framing accordingly, while keeping megafloods as end-members that clarify the limits of the exceedance-probability formulation. We also added a brief mention of ancient megafloods (and analogous events on other planetary bodies) as boundary cases that motivate this second end-member. These changes are reflected in Section 2 and in the challenges and outlook section (Section 5).

Anthropocentric floods – defined as “physical floods deemed desirable or undesirable”

The complexity of human decisions that have to be made in regard to flooding are well illustrated by the very recent paper on where formal rule curves intersect with human discretionary judgment in regard to reservoir operations relative to flood risk (Gautam et al., 2025). It would be well to check how the demarcation of “anthropocentric floods” fits with this aspect of flood hydrology.

We agree with Prof Baker and thank him for bringing this paper to our attention. This is a very interesting point and one which we would like to investigate in further depth. However, to do this we would like to elicit the assistance of behavioral scientists to conduct surveys and see how conceptions of desirability are utilized by individuals in these decision-making roles. In this way, we see this comment as a genuine point of novel investigation which should be done within our future work.

Normativity

Normativity is only one approach to the general issue of values. An alternative approach would be to consider the complete axiology of the situation (e.g., Baker, 1998).

As noted in the discussion of “interventional ambiguity” the focus of this manuscript is what is desirable or undesirable in regard to the physical phenomenon of (hydroclimatic riverine) flooding. It is claimed (lines 363-364.) that, “...normativity is specifically regarding adopting a notion of desirability for the presence of water at the identified time and location.” In other words, what an individual wants or wishes for (desires) as a course action is to be specified by what ought to be (normativity) in regard to those desires. This approach assumes a standard (a norm) to be applied to those desires that might overcome the problem of deciding what action to take (the “interventional ambiguity”).

We thank Prof. Baker for this comment, and agree that one of the focuses of this paper is to discuss the normativity involved in adopting a conception of desirability for the presence of water. This was done in relation to demarcating anthropocentric floods from physical floods as well as the emergent feature of interventional ambiguity, which the Professor has highlighted here. However, an additional point of this discussion is on the normativity involved in identifying physical floods as well, where an individual must normatively define the relevance parameters and rarity threshold.

Desirability in regard to flooding, "...is always to be done in a decision-making context where a level of normative uncertainty is present" (lines 431-432). Normative uncertainty involves, "...what should be the case..." (line 435). There is going to be uncertainty, i.e., doubt or unsureness, about the desirability of flooding. This is the classical concern about making a value judgment as to what action to take. One might like a sound basis for the action that might be viewed as being without the possibility of error.

The problem of doubt (uncertainty) in regard to what action to take, i.e., what would be most desirable to do, is an age-old. If one could have an ideal sense, without error, of what would happen by taking a particular action, then that potential action could be viewed as desirable or undesirable. In the 19th century the famous English mathematician William Kingdom Clifford addressed this problem as follows (see Clifford, 1946):

We agree, the problem of doubt and error is a problem discussed for a very long time. However, aside from our doubt in knowing what will happen in light of certain actions taking place (this doubt being empirical uncertainty), there is also uncertainty regarding what *should* take place (where this doubt would be normative uncertainty). Thus, it is not only about concerns regarding the error in decision-making, where the outcome was not the intention, but also in what one wishes that outcome to be or ought to be.

Remember, then, that scientific thought is the guide of action; that the truth at which it arrives is not that which we can ideally contemplate without error, but that which we may act upon without fear.

So, how does one find, "...that which we can act upon without fear." In other words, how can the doubt (uncertainty) be removed? Doubt is overcome by belief, which is the antonym of doubt. Belief will be the accepting of something, in this case the desirability that a particular action will be the correct one, or at least it is proving to be worthy of overcoming the doubt (uncertainty) about that action.

However, there are multiple methods for the fixing of belief (Peirce, 1877) - the most effective of these for the long term being that of science. Here "science" does not consist of a demarcation

criterion for what ought to be done, but rather science is a continuing pursuit of truth by following the consequences of and learning from the actions that are taken.

We thank Prof. Baker for these comments. We agree that knowledge produced by a scientific process is often used as the basis for guiding action. Through investigation, science can provide a better understanding of the world and how/why it operates in a certain way. This allows for a certain level of agreement between individuals on “the facts of the matter” when making decisions. In this way, science is an excellent tool for addressing the “empirical uncertainty” present within a decision-making context, reducing our uncertainty about *the actual state of the world*. However, science does not provide us insight in addressing normative uncertainty, the uncertainty about *the way things ought to be*. We have made efforts within the manuscript to stress this point, highlighting the value-ladenness of science in addressing the normative uncertainty. There is an explicit discussion of these two forms of uncertainty within Section. 4 when we begin to argue for interventional ambiguity as an emergent feature of our framework.

Similarly the meaning of any concept, e.g., “flood,” is not something that is best set by a standard. Instead, meaning consists of the effects that conceivably follow as the practical consequences of that concept’s operation in the world (Peirce, 1878). This can lead to an axiology in regard to human aspects of flooding that derives from the philosophy of pragmatism (Baker, 1994, 1998, 2007).

Philosophical Pragmatism

The philosophical application of normativity seems directed at the goal of “flood-risk management before...problems arise...” (lines 529-531). This is an engineering approach rather than a scientific one. The relevant point here is about pragmatism as a philosophy relevant to the issue of flood risk (see Baker, 2007). The philosophical pragmatist seeks clarity in the concept “flood,” but that clarity cannot be achieved in advance of the application of the concept. This philosophical perspective has been around for at least 150 years (Peirce, 1877, 1878), and it is worthy of more consideration for the kinds of issues addressed in this manuscript.

We thank Prof. Baker for this comment and directing us towards Peirce’s work on pragmatism, a very rich discussion within philosophy of science, as well as their work on pragmatist approaches. However, to avoid confusion, we want to clarify one point on our application of normativity in this context. While one application of normativity can be seen, as Prof. Baker has stated, to be “directed at the goal of ‘flood-risk management before...problems arise...’”, this is not the only application. The influence of normativity is first recognized within our framework when an individual must identify the relevance parameters and the rarity threshold. The use of normativity in this aspect of the framework is not specifically in relation to the management of floods, in an engineering sense, but is specifically focused on purely identifying when a physical

flood occurs. Defining these parameters and the threshold using a conception of what ought to be has significant impacts for the performance of science and the scientific community's understanding of why certain events are classified/identified as a flood and why others are not. We have attempted to revise the manuscript, across various sections, to emphasize this point that our discussion goes beyond questions of management and the engineering of these relevant systems.

CONCLUSION

The above comments could lead to a paper providing a completely different answer to the question that is expressed in the title of this manuscript. Rather than doing that, however, the easiest course for revising the present manuscript might be to acknowledge the various limitations and caveats on the “conceptual paradigm” that have been noted above. This revision should also emphasize the very limited applicability of that paradigm to the original question, restricting the answer to “hydrometeorological riverine floods” and their various consequences for human concerns.

This inclusion of limitations on the arguments presented in the paper might also be extended to the role of normativity, since its inclusion does not provide a complete axiology (science of values). Nevertheless, whether the authors wish to consider in regard to a more flexible axiology that would embrace a pragmatic approach (e.g., Baker, 2007) is up to them. Nearly all philosophical arguments can be rationalized, and it can always be interesting to see them presented.

We agree with Prof. Baker. In the revision, we added a dedicated section that explicitly lists and discusses the key limitations, caveats, and scope of applicability of the proposed conceptual paradigm (this can be seen in Sec. 5 Potential challenges and outlook). We also revised the surrounding text (Abstract/Introduction/subsequent discussions) to make the manuscript clearer. In sum, we would like to thank Prof. Baker for the indepth comments and reflections on the topics we are discussing in this manuscript. We feel that the manuscript has improved dramatically from the original version, and the inclusion of these additional thoughts/concerns of Prof. Baker have been incredibly helpful.

References Cited in This Review

Baker, V.R., 1994, Geomorphological understanding of floods: *Geomorphology*, v. 10, p. 139

156.

Baker, V.R., 1998, Hydrological understanding and societal action: *Journal of the American Water Resources Association*, v. 34, no. 4, p. 819-825.

Baker, V.R., 2007, Flood hazard science, policy and values: A pragmatist stance: *Technology in Society*, v. 29, p. 161-168.

Baker, V.R., 2020, Global megaflood paleohydrology, in Herget, J. and Fontana, A., editors, *Palaeohydrology. Geography of the Physical Environment*: Springer, Heidelberg, p. 3-28.

Baker, V.R., and Carling, P.A., 2022, Global Late Quaternary Megafloods, in Shroder, J.J.F. (Ed.), *Treatise on Geomorphology*, 2nd Edition, vol. 6. Elsevier, Academic Press, p. 832-840.

Baker, V.R., and Milton, D.J., 1974, Erosion by Catastrophic Floods on Mars and Earth: *Icarus*, v. 23, p. 27-41.

Burr, D.M., Carling, P.A., and Baker, V.R., editors, 2009, *Megaflooding on Earth and Mars*: Cambridge University Press, Cambridge, 319 p.

Clifford WK. *The common sense of the exact sciences*. New York: Alfred A. Knopf; 1946. (Reprinted from Clifford, William K. 1885. *The Common Sense of the Exact Sciences*, completed by K. Pearson. London: Kegan, Paul, Trench, and Co.)

Gautam, S., Park, S., Yu, D.J., Garcia, M., Sivapalan, M., and Shin, H.C., 2025, Homo Juridicus, Homo Heuristicus, and Homo Anticipans: A Sociohydrological Study of Operator Behavior and Flood-Drought Tradeoffs in Reservoirs: *Water Resources Research*, v. 61, issue 11, e2024WR039447 <https://doi.org/10.1029/2024WR039447>

Peirce, C. S., 1877, The fixation of belief: *Popular Science Monthly*, v. 12, p. 1-15.

Peirce, C.S., 1878, How to make our ideas clear: *Popular Science Monthly*, v. 12, p. 388-410.

Reviewer #3 (Remarks to the Author):

This is a great paper that brings much needed clarity to the concept of "flood". It is well written and well-argued, and is accessible to those who might not have technical expertise in hydrology. I also want to applaud the authors for bringing philosophical analysis to a challenge in hydrology and for presenting a strong example of how these two seemingly disparate disciplines can benefit from one another's perspectives. I recommend publication, as I think the content of the paper add value to the community.

But, I do suggest two minor revisions, one of which is not necessary to include, but I think would articulate the additional impact/importance of the discussion.

1) (Necessary) One thing that surprised me in reading this was the lack of explicit connection between normativity and values. The values in science literature is extensive, and essentially, what this paper is getting at is that decisions made about how to define a "flood" is dependent on context, and more specifically, the values that are operating in that context. While I don't mind the use of "normativity" (this is a unique dimension this discussion adds, which I appreciate) as opposed to explicitly talking about "value-ladenness" I do think there needs to be a note early on to the effect of "By normativity, we refer to the evaluative judgments that underly determinations of what ought to be—judgments that necessarily involve values about desirable states of affairs in certain locations at certain points in time". And then include a reference early on to Elliott.

You should include this in the third introduction paragraph—somewhere in this portion" Here, we argue that utilizing normative aspects, adopting a notion of the way things ought to or should be [4], is necessary for any conceptualization of flood, however, this currently remains tacit. Therefore, any comprehensive demarcation framework for the flood phenomenon must explicitly recognize the role of this normativity." This is essentially a statement about the value-ladenness of the way different researchers/stakeholders etc. might conceptualize a "flood". The authors need to make this connection and reference some overview of this history of the discussions of how values impact scientific practice from the philosophy of science.

We completely agree with the reviewer's comment here and want to thank them for pointing this out. This connection to the literature on values, within the philosophy of science literature, should be made more explicit. We have added a sentence, as suggested, in the third paragraph, which better outlines the connection between normativity, evaluative judgements, and value-ladenness.

2. (Unnecessary) The conceptual clarity that this paper provides does some critical work for making flood research more usable/actionable: a) when researchers and stakeholders engage in

the explicit identification of normative commitments (desirability, thresholds, etc.) to build the definition of a flood for a specific context, the resulting research presumably becomes more salient (relevant) to specific decision contexts, even with the potential for interventional ambiguity. Communities are better able to assess, when these components of the conceptual space are explicitly articulated, to better assess whether the implicit values in a flood model or dataset align with their own values and priorities, and it makes value deliberation and assessment of the usability of research easier. In this way it also has impacts for legitimacy of research by making value explicit and subject to democratic deliberation. Values, or normativity, as part of the conceptual process, are no longer "smuggled in" as objective technical choices, so these normative components can be objects of deliberation, making the evaluation of the legitimacy (component of usability) easier.

You could include something in the conclusion like "By explicating the normative components that fix flood concepts in specific contexts, researchers can produce work that is not only scientifically rigorous but also more transparent about its value assumptions. This transparency can serve multiple purposes: it increased the salience of researcher to particular decision contexts by making clear which values the research embeds; it enhances legitimacy by allowing stakeholders to evaluate whether those values align with community priorities and allow them to function as objects of democratic deliberation. Conceptual clarity in hydrology about the value-ladenness of how we define a flood is not just philosophically interesting, but a step towards making research more usable and actionable for responding to climate hazards". And I would cite something like "Usability of climate information: Toward a new scientific framework".

We thank the reviewer for bringing this to our attention and completely agree with this point. The conclusions of this work have clear connections to these topics that have become increasingly prevalent within the philosophical literature and, subsequently, there are further impacts resulting from our proposed demarcation framework. We included a discussion of this within the final paragraph of our new section, "5. Potential challenges and outlook". In this paragraph we were able to provide references to a number of key papers within the recent literature in philosophy of (climate) science, including the paper on usability, highlighted by the reviewer. We hope that this can build further connections between our work in this perspective piece and related issues within the philosophical literature, where we can draw further attention to these important discussions.

Ultimately, we would like to thank this reviewer for providing their perspective and giving us these helpful comments. The comments provided by this reviewer have increased the salience of the connection between values and normativity within this work, generally, as well as providing an earlier reference to this point. Additionally, the comments greatly improved the connection between this paper and additional discussions/topics within the philosophy of science, allowing

new readers to think about the implications of this work beyond what has been specifically discussed in this paper.